# SASSHA: Sharpness-aware Adaptive Second-order Optimization with Stable Hessian Approximation

Dahun Shin [* 1]   Dongyeop Lee [* 1]   Jinseok Chung [1]   Namhoon Lee [1]

## Abstract

Approximate second-order optimization methods often exhibit poorer generalization compared to first-order approaches. In this work, we look into this issue through the lens of the loss landscape and find that existing second-order methods tend to converge to sharper minima compared to SGD. In response, we propose SASSHA, a novel second-order method designed to enhance generalization by explicitly reducing sharpness of the solution, while stabilizing the computation of approximate Hessians along the optimization trajectory. In fact, this sharpness minimization scheme is crafted also to accommodate lazy Hessian updates, so as to secure efficiency besides flatness. To validate its effectiveness, we conduct a wide range of standard deep learning experiments where SASSHA demonstrates its outstanding generalization performance that is comparable to, and mostly better than, other methods. We provide a comprehensive set of analyses including convergence, robustness, stability, efficiency, and cost.

## 1. Introduction

Approximate second-order methods have recently gained a surge of interest due to their potential to accelerate the large-scale training process with minimal computational and memory overhead (Yao et al., 2021; Liu et al., 2024; Gupta et al., 2018). However, studies also suggest that these methods may undermine generalization, trying to identify underlying factors behind this loss (Wilson et al., 2017; Zhou et al., 2020; Zou et al., 2022). For instance, Amari et al. (2021) shows that preconditioning hinders achieving the optimal bias for population risk, and Wadia et al. (2021) points to negative effect of whitening data.

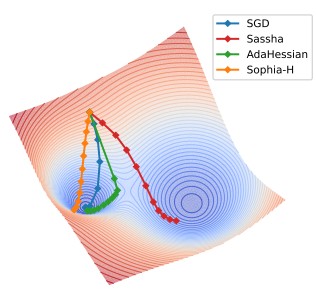

Figure 1. Motivating toy example (a mixture of bivariate Gaussian densities). SASSHA converges to a flat minimum unlike others.

While the precise understanding is still under investigation, many studies have suggested a strong correlation between the flatness of minima and their generalization capabilities (Keskar et al., 2017), spurring the development of optimization techniques aimed at inducing flat minima (Chaudhari et al., 2017; Izmailov et al., 2018; Foret et al., 2021; Orvieto et al., 2022). Inspired by this, we raise an important question in this work: what type of minima do second-order methods converge to, and is there any potential for improving their generalization performance based on that?

To answer these questions, we first measure the sharpness of different second-order methods using diverse metrics, suggesting that they converge to significantly sharper minima compared to stochastic gradient descent (SGD). Then, we propose SASSHA—**S**harpness-aware **A**daptive **S**econd-order optimization with **S**table **H**essian **A**pproximation—designed to enhance the generalization of approximate second-order methods by explicitly reducing sharpness (see Figure 1 for the basic results).

SASSHA incorporates a sharpness minimization scheme similar to SAM (Foret et al., 2021) into the second-order optimization framework, in which the Hessian diagonal is estimated. Such estimates, however, can become numerically unstable when enforcing the sharpness reduction process. To increase stability while preserving the benefits of reduced sharpness, we make a series of well-engineered design choices based on principles studied in the literature. This not only smoothly adjusts underestimated curvature, but also enables efficient reuse of previously computed Hes-

*Equal contribution   [1]POSTECH. Correspondence to: Dahun Shin <dahunshin@postech.ac.kr>, Dongyeop Lee <dylee23@postech.ac.kr>.

*Proceedings of the 42nd International Conference on Machine Learning*, Vancouver, Canada. PMLR 267, 2025. Copyright 2025 by the author(s).

sians, resulting in a stable and efficient algorithm.

We extensively evaluate the effectiveness of SASSHA across diverse vision and natural language tasks. Our results reveal that SASSHA consistently achieves flatter minima and attains stronger generalization performance, all compared to existing practical second-order methods, and interestingly, to first-order methods including SGD, AdamW, and SAM. Furthermore, we provide an array of additional analyses to comprehensively study SASSHA including convergence, robustness, stability, efficiency, and cost.

## 2. Related Works

**Second-order optimization for deep learning.** First-order methods such as SGD are popular optimization methods for deep learning due to their low per-iteration cost and good generalization performance (Hardt et al., 2016). However, these methods have two major drawbacks: slow convergence under ill-conditioned landscapes and high sensitivity to hyperparameter choices such as learning rate (Demeniconi & Chawla, 2020). Adaptive methods (Duchi et al., 2011; Hinton et al., 2012; Kingma & Ba, 2015) propose using empirical Fisher-type preconditioning to alleviate these issues, though recent studies suggest their insufficiency to do so (Kunstner et al., 2019). This has led to recent interest in developing approximate second-order methods such as Hessian-Free Inexact Newton methods (Martens et al., 2010; Kiros, 2013), stochastic quasi-Newton methods (Byrd et al., 2016; Gower et al., 2016), Gauss-Newton methods (Schraudolph, 2002; Botev et al., 2017), natural gradient methods (Amari et al., 2000), and Kronecker-factored approximations (Martens & Grosse, 2015; Goldfarb et al., 2020). However, these approaches still incur non-trivial memory and computational costs, or are difficult to parallelize, limiting their applicability to large-scale problems such as deep learning. This has driven growing interest in developing more scalable and efficient second-order approaches, particularly through diagonal scaling methods (Bottou et al., 2018; Yao et al., 2021; Liu et al., 2024), to better accommodate large-scale deep learning scenarios.

**Sharpness minimization for generalization.** The relationship between the geometry of the loss landscape and the generalization ability of neural networks was first discussed in the work of Hochreiter & Schmidhuber (1994), and the interest in this subject has persisted over time. Expanding on this foundation, subsequent studies have explored the impact of flat regions on generalization both empirically and theoretically (Hochreiter & Schmidhuber, 1997; Keskar et al., 2017; Dziugaite & Roy, 2017; Neyshabur et al., 2017; Dinh et al., 2017; Jiang et al., 2020). Motivated by this, various approaches have been proposed to achieve flat minima such as regularizing local entropy (Chaudhari et al.,

2017), averaging model weights (Izmailov et al., 2018), explicitly regularizing sharpness by solving a min-max problem (Foret et al., 2021), and injecting anti-correlated noise (Orvieto et al., 2022), to name a few. In particular, the sharpness-aware minimization (SAM) (Foret et al., 2021) has attracted significant attention for its strong generalization performance across various domains (Chen et al., 2022; Bahri et al., 2022; Qu et al., 2022) and its robustness to label noise (Baek et al., 2024). Nevertheless, to our knowledge, the sharpness minimization scheme has not been studied to enable second-order methods to find flat minima as of yet.

## 3. Practical Second-order Optimizers Converge to Sharp Minima

In this section, we investigate the sharpness of minima obtained by approximate second-order methods and their generalization properties. We posit that poor generalization of second-order methods reported in the literature (Amari et al., 2021; Wadia et al., 2021) can potentially be attributed to sharpness of their solutions.

We employ four metrics frequently used in the literature: maximum eigenvalue of the Hessian, the trace of Hessian, gradient-direction sharpness, and average sharpness (Hochreiter & Schmidhuber, 1997; Jastrzębski et al., 2018; Xie et al., 2020; Du et al., 2022b; Chen et al., 2022). The first two, denoted as $\lambda_{\max}(H)$ and $\mathrm{tr}(H)$, are often used as standard mathematical measures for the worst-case and the average curvature computed using the power iteration method and the Hutchinson trace estimation, respectively. The other two measures, $\delta L_{\mathrm{grad}}$ and $\delta L_{\mathrm{avg}}$, assess sharpness based on the loss difference under perturbations. $\delta L_{\mathrm{grad}}$ evaluates sharpness in the gradient direction and is computed as $L(x^\star + \rho \nabla L(x^\star)/\|\nabla L(x^\star)\|) - L(x^\star)$. $\delta L_{\mathrm{avg}}$ computes the average loss difference over Gaussian random perturbations, expressed as $\mathbb{E}_{z\sim\mathcal{N}(0,1)}[L(x^\star + \rho z/\|z\|) - L(x^\star)]$. Here we choose $\rho = 0.1$ for the scale of the perturbation.

With these, we measure the sharpness of the minima found by three approximate second-order methods designed for deep learning; Sophia-H (Liu et al., 2024), AdaHessian (Yao et al., 2021), and Shampoo (Gupta et al., 2018), and compare them with SASSHA as well as SGD for reference. We also compute the validation loss and accuracy to see any correlation between sharpness and generalization of these solutions. The results are presented in Table 1.

We observe that existing second-order methods produce solutions with significantly higher sharpness compared to SASSHA in all sharpness metrics, which also correlates well with their generalization. We also provide a visualization of the loss landscape for the found solutions, where we find that the solutions obtained by second-order methods are indeed much sharper than that of SASSHA (Figure 2).

*Table 1.* Sharpness measurements of the solutions found by different optimizers and their generalization for ResNet-32 on CIFAR-100. Approximate second-order methods tend to yield highly sharp solutions and poor generalization compared to SASSHA. We provide more results for other workloads in Appendix A where the same trend holds.

| | Sharpness | | | | Generalization | |
|---|---|---|---|---|---|---|
| | $\lambda_{max}(H)$ | $\mathrm{tr}(H)_{\times 10^3}$ | $\delta L_{\mathrm{grad}}$ | $\delta L_{\mathrm{avg} \times 10^{-3}}$ | $L_{\mathrm{val}}$ | $\mathrm{Acc}_{\mathrm{val}}$ (%) |
| SGD | $265_{\pm 25.00}$ | $7.290_{\pm 0.300}$ | $0.703_{\pm 0.132}$ | $1.310_{\pm 1.030}$ | $1.260_{\pm 0.001}$ | $69.32_{\pm 0.19}$ |
| AdaHessian | $11992_{\pm 5779}$ | $46.94_{\pm 17.60}$ | $4.119_{\pm 1.136}$ | $12.50_{\pm 6.080}$ | $1.377_{\pm 0.070}$ | $68.06_{\pm 0.22}$ |
| Sophia-H | $22797_{\pm 10857}$ | $68.15_{\pm 20.19}$ | $8.130_{\pm 3.082}$ | $19.19_{\pm 6.380}$ | $1.463_{\pm 0.022}$ | $67.76_{\pm 0.37}$ |
| Shampoo | $436374_{\pm 9017}$ | $6823_{\pm 664.7}$ | $73.27_{\pm 12.51}$ | $49307489_{\pm 56979794}$ | $1.386_{\pm 0.010}$ | $64.08_{\pm 0.46}$ |
| SASSHA | $107_{\pm 40.00}$ | $1.870_{\pm 0.650}$ | $0.238_{\pm 0.088}$ | $0.650_{\pm 0.860}$ | $0.961_{\pm 0.005}$ | $72.14_{\pm 0.16}$ |

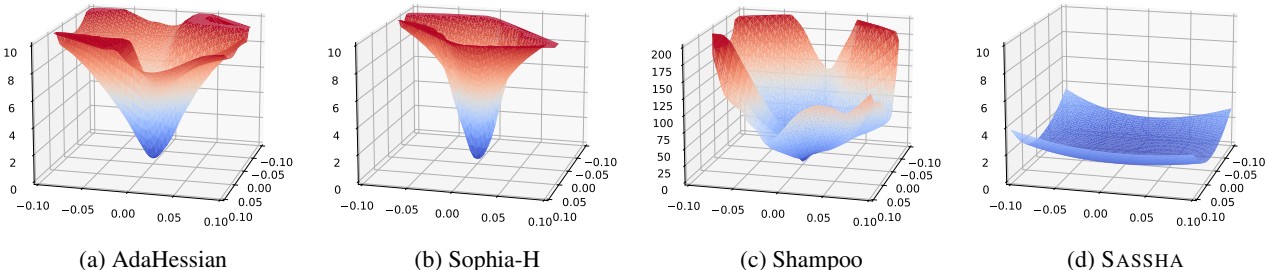

(a) AdaHessian     (b) Sophia-H     (c) Shampoo     (d) SASSHA

*Figure 2.* Visualization of the found solutions along the directions of the dominant eigenvectors.

## 4. Method

In the previous section, we observe that the generalization performance of approximate second-order algorithms anti-correlates with the sharpness of their solutions. Based on this, we introduce SASSHA—a novel adaptive second-order method designed to improve generalization by reducing sharpness without adversely impacting the Hessian.

### 4.1. Sharpness-aware Second-order Optimization

We consider a min-max problem, similar to Keskar et al. (2017); Foret et al. (2021), to minimize sharpness. This is defined as minimizing the objective $f$ within the entire $\rho$-ball neighborhood:

$$\min_{x \in \mathbb{R}^d} \max_{\|\epsilon\|_2 \leq \rho} f(x + \epsilon), \qquad (1)$$

Based on this, we construct our sharpness minimization technique for second-order optimization as follows. We first follow a similar procedure as Foret et al. (2021) by solving for $\epsilon$ on the first-order approximation of the objective, which exactly solves the dual norm problem as follows:

$$\epsilon_t^\star = \arg\max_{\|\epsilon\|_2 \leq \rho} f(x_t) + \epsilon^\top \nabla f(x_t) = \rho \frac{\nabla f(x_t)}{\|\nabla f(x_t)\|_2}. \quad (2)$$

We plug this back to yield the following perturbed objective

function:

$$\tilde{f}_t(x) := f\left(x + \rho \frac{\nabla f(x_t)}{\|\nabla f(x_t)\|_2}\right), \qquad (3)$$

which shifts the point of the approximately highest function value within the neighborhood to the current iterate.

With this sharpness-penalized objective, we proceed to make a second-order Taylor approximation:

$$x_{t+1} = \arg\min_x \tilde{f}_t(x_t) + \nabla \tilde{f}_t(x_t)^\top (x - x_t)$$
$$+ (x - x_t)^\top \tilde{H}_t(x_t)(x - x_t),$$

where $\tilde{H}_t$ denotes the Hessian of $\tilde{f}_t$. Using the first-order optimality condition, we derive the basis update rule for our sharpness-aware second-order optimization:

$$x_{t+1} = x_t - \tilde{H}_t(x_t)^{-1} \nabla \tilde{f}_t(x_t)$$
$$= x_t - H(x_t + \epsilon_t^\star)^{-1} \nabla f(x_t + \epsilon_t^\star), \qquad (4)$$

where $H$ denotes the Hessian of the original objective $f$.

Practical second-order methods must rely on approximately estimated Hessians (*i.e.*, $H \to \widehat{H}$) since the exact computation is prohibitively expensive for large-scale problems. We choose to employ the diagonal approximation via Hutchinson's method. However, as we will show in our analysis (Section 6.2), we find that these estimates can become numerically unstable during the sharpness reduction process,

as it penalizes Hessian entries close to zero. This can lead to fatal underestimation of the diagonal Hessian compared to scenarios without sharpness minimization, significantly disrupting training. We propose a stable Hessian approximation to address these issues in the following sections.

### 4.2. Improving Stability

**Alleviating divergence.** Approximate second-order methods can yield overly large steps when their diagonal Hessian estimations underestimate the curvature (Dauphin et al., 2015). However, this instability seems to be more present under sharpness minimization, presumably due to smaller top Hessian eigenvalue $\lambda_1$ (Agarwala & Dauphin, 2023; Shin et al., 2025) yielding smaller diagonal entries:

$$|H_{ii}| = |\mathbf{e}_i^\top H \mathbf{e}_i| \leq \max_{\|v\|_2 = 1} |v^\top H v| = |\lambda_1|,$$

where $\mathbf{e}_i$ is the $i^{\text{th}}$ standard basis vector. This tendency toward zero intensifies numerical instability during Hessian inversion, increasing the risk of training failures.

A conventional approach to mitigating this involves damping or clipping, which, while stabilizing it reasonably well, requires carefully tuning their additional hyperparameters. Instead, we propose square rooting the Hessian (*i.e.*, $|\widehat{H}|^{1/2}$), which effectively mitigates instability and allows improved generalization performance over other alternatives without additional hyperparameters. We present empirical validation of this in Section 6.2 and Appendix F.1.

Its benefits can be understood from two perspectives. First, the square root preserves the relative scale between each element of the Hessian while smoothly increasing the magnitude of the near-zero diagonal Hessian entries in the denominator (*i.e.*, $h < \sqrt{h}$ if $0 < h < 1$). This property is particularly valuable when the sharpness minimization is underway, where the overall Hessian values tend to be small. In such cases, even small differences between Hessian elements may carry nontrivial curvature information. Square-rooting can help retain this relative structure while also mitigating numerical instability caused by underestimated curvature. In contrast, both damping and clipping operate by entirely shifting or abruptly replacing Hessian values based on a predefined and fixed threshold criterion. As a result, when the Hessian is generally small due to sharpness minimization, informative dimensions may fall below the threshold, removing potentially critical variations and hence deteriorating the quality of preconditioning. This behavior can also make the method more sensitive to the choice of the threshold hyperparameter.

Second, it can further be interpreted as a geometric interpolation between the identity matrix $I$ and the preconditioning matrix $|\widehat{H}|^\alpha$, which, as theoretically analyzed in (Amari et al., 2021), provide a natural mechanism for balancing between bias and variance of the population risk, thereby improving generalization. We specifically adopt $\alpha = 1/2$ (i.e., square root), as it has consistently demonstrated robust performance across various scenarios (Amari et al., 2021; Kingma & Ba, 2015).

**Avoiding critical points.** A well-known limitation of applying second-order optimization to deep learning objectives is the risk of convergence to saddle points or local maxima. To mitigate this, we attend to the prior works of Becker et al. (1988); Yao et al. (2021) and employ the absolute function to conservatively adjust the negative entries of the diagonal Hessian to be positive, *i.e.*

$$|\widehat{H}| := \sum_{i=1}^{d} |\widehat{H}_{ii}| \mathbf{e}_i \mathbf{e}_i^\top \tag{5}$$

where $\widehat{H}_{ii}$ and $\mathbf{e}_i$ are the $i^{\text{th}}$ diagonal entry of the approximate diagonal Hessian and the $i^{\text{th}}$ standard basis vector, respectively. Here, the basic idea is to maintain the directionality of the gradient by flipping the sign of the negative entries in the Hessian, preserving the original magnitude of its values. This allows our method not only to take descent steps along directions of originally negative curvature, but also to preserve the relative scale specifically among the negative elements of the Hessian. As a result, it mitigates the risk of convergence to the undesired critical points. We empirically validate the effectiveness of this approach in Appendix F.2.

### 4.3. Improving Efficiency via Lazy Hessian Update

While the diagonal Hessian approximation can significantly reduce computations, it still requires at least twice as much backpropagation compared to first-order methods. Here we attempt to further alleviate this by lazily computing the Hessian every $k$ steps:

$$D_t = \begin{cases} \beta_2 D_{t-1} + (1 - \beta_2)|\widehat{H}(x_t + \epsilon_t^\star)| & \texttt{if } t \bmod k = 1 \\ D_{t-1} & \texttt{otherwise} \end{cases},$$

where $D_t$ and $\beta_2$ are the moving average of the Hessian and its hyperparameter, respectively. This reduces the overhead from additional Hessian computation by $1/k$. We set $k = 10$ for all experiments in this work unless stated otherwise.

However, extensive Hessian reusing will lead to significant performance degradation since it would no longer accurately reflect the current curvature (Doikov et al., 2023). Interestingly, SASSHA is quite resilient against prolonged reusing, keeping its performance relatively high over longer Hessian reusing compared to other approximate second-order methods. Our investigation reveals that along the trajectory of

SASSHA, the Hessian tends to change less frequently than existing alternatives. We hypothesize that the introduction of sharpness minimization plays an integral role in this phenomenon by biasing the optimization path toward regions with lower curvature change, allowing the prior Hessian to remain relevant over more extended steps. We provide a detailed analysis of the lazy Hessian updates in Section 6.3.

### 4.4. Algorithm

The exact steps of SASSHA is outlined in Algorithm 1. We also compare SASSHA with other adaptive and second-order methods in detail in Appendix B, where one can see the exact differences between these sophisticated methods.

### 4.5. Convergence Analysis

In this section, we present a standard convergence analysis of SASSHA under the following assumptions.

**Assumption 4.1.** The function $f$ is bounded from below, i.e., $f^* := \inf_x f(x) > -\infty$.

**Assumption 4.2.** The function $f$ is twice differentiable, convex, and $\beta$-smooth. That is, $0 \preceq \nabla^2 f \preceq \beta$.

**Assumption 4.3.** *The gradient $\nabla f(x_t)$ is nonzero for a finite number of iterations, i.e., $\nabla f(x_t) \neq 0$ for all $t \in \{1, 2, \ldots, n\}$.*

Under these assumptions, we derive a descent inequality for $f(x_t)$ by leveraging Adam-like proof techniques from Li et al. (2023) to handle the diagonal Hessian and employing smoothness-based bounds to account for the perturbation step based on analyses of Khanh et al. (2024). Now we give the convergence results as follows:

**Theorem 4.4.** *Under Assumptions 4.1-4.3, given any initial point $x_0 \in \mathbb{R}^d$, let $\{x_t\}$ be generated by the update rule SASSHA Equation (8) with step sizes $\eta_t$ and perturbation radii $\rho_t$ satisfying $\sum_{t=1}^{\infty} \eta_t = \infty$, $\sum_{t=1}^{\infty} \eta_t^2 < \infty$, $\sum_{t=1}^{\infty} \rho_t^2 \eta_t < \infty$. Then, we have $\liminf_{t \to \infty} \|\nabla f(x_t)\| = 0$.*

This preliminary result indicates that any limit point of SASSHA is a stationary point of $f$, ensuring progress towards optimal solutions. We refer to Appendix C for the full proof details.

### 4.6. Flatness Analysis

To understand how SASSHA can end up in flatter minima as observed in Section 3, we attend to linear stability analysis. Originally developed to explain similar behavior in SGD (Wu et al., 2018) and also later extended to SAM (Shin et al., 2025), this framework suggests that an optimizer does not settle in every minimum it approaches, but instead escapes unless it encounters one that satisfies specific stability conditions—conditions that can vary between optimizers. Based

---

**Algorithm 1** SASSHA algorithm

---
1: **Input:** Initial parameter $x_0$, learning rate $\{\eta_t\}$, moving average parameters $\beta_1, \beta_2$, Hessian update interval $k$, weight decay parameter $\lambda$
2: Set $m_{-1} = 0$, $D_{-1} = 0$
3: **for** $t = 1$ **to** $T$ **do**
4:     $g_t = \nabla f_\mathcal{B}(x_t)$
5:     $\epsilon_t^\star = \rho g_t / \|g_t\|_2$
6:     $\tilde{g}_t = \nabla f_\mathcal{B}(x_t + \epsilon_t^\star)$
7:     $m_t = \beta_1 m_{t-1} + (1 - \beta_1)\tilde{g}_t$
8:     $\overline{m}_t = m_t / (1 - \beta_1^t)$
9:     **if** $t \bmod k = 1$ **then**
10:        $\tilde{H}_t = \widehat{H}(x_t + \epsilon_t^\star)$       ▷ Section 4.1
11:        $D_t = \beta_2 D_{t-1} + (1 - \beta_2)|\tilde{H}_t|$
12:        $\overline{D}_t = \sqrt{D_t / (1 - \beta_2^t)}$     ▷ Section 4.2
13:     **else**
14:        $\overline{D}_t = \overline{D}_{t-1}$          ▷ Section 4.3
15:     **end if**
16:     $x_{t+1} = x_t - \eta_t \overline{D}_t^{-1} \overline{m}_t - \eta_t \lambda x_t$
17: **end for**

---

on this, we demonstrate that SASSHA requires a minimum to possess a certain level of flatness and Hessian uniformity to settle in it, whereas approximate second-order optimizers do not necessarily require such restriction, thus allowing it to stay in much sharper minima.

Consider a general optimizer $x_{t+1} = x_t - G(x_t; \xi_t)$ with randomness induced by i.i.d. variables $\xi_t$ that are independent from the iterate $x_t$. Also, let us assume that the minima possess a fixed point $x^\star$ such that $\nabla f_\xi(x^\star) = 0$ for any $\xi$.

With these, we define the linear stability of the fixed point $x^\star$ as follows:

**Definition 4.5.** (Linear stability). A fixed point $x^\star$ is *linearly stable* for the optimizer $G$ if there exists a constant $C$ such that

$$\mathbb{E}\big[\|\hat{x}_t - x^\star\|^2\big] \leq C\|\hat{x}_0 - x^\star\|^2, \text{ for all } t > 0$$

under the linearized dynamics near $x^\star$: $\hat{x}_{t+1} = \hat{x}_t - \nabla G(x^\star)(\hat{x}_t - x^\star)$, *i.e.*, when it does not deviate infinitely far from $x^\star$.

Here, this linear dynamic arises when $x_t$ has approached sufficiently close to $x^\star$ such that the loss becomes approximately quadratic.

Under this framework, we present the necessary conditions of the linearly stable minima of SASSHA in the following corollary.

**Corollary 4.6.** *Assume without loss of generality that $x^\star = 0$. Then, the linearly stable fixed point $x^\star$ of SASSHA satisfy*

*the following necessary conditions:*

$$0 \leq a(1 + \rho a) \leq \frac{2\epsilon}{\eta}, \quad 0 \leq s_2^2 \leq \frac{\epsilon^2}{\eta^2 - 2\eta\rho\epsilon},$$

$$0 \leq s_3^2 \leq \frac{\epsilon^2}{2\eta^2\rho}, \quad 0 \leq s_4^2 \leq \frac{\epsilon^2}{\eta^2\rho^2}, \quad (6)$$

*where $a = \lambda_{max}(\mathbb{E}[H_\xi])$ and $s_k = \lambda_{max}((\mathbb{E}[H_\xi^k] - \mathbb{E}[H_\xi]^k)^{1/k})$ are the sharpness and the non-uniformity of the stochastic Hessian measured with the $k$-th moment, respectively.*

A detailed proof is provided in Appendix D.

These results indicate that SASSHA escapes from minima unless they satisfies both low sharpness and uniformly distributed Hessian moments, with the conditions becoming much stricter with larger $\rho$ and $\eta$. In comparison, standard approximate second-order methods of the form $x_{t+1} = x_t - \eta P_t \nabla f(x_t)$ with $P \approx H^{-1}$ remain stable without such conditions, as shown by Wu et al. (2018):

$$\lambda_{\max}(P^{-1}H) \approx 1 \leq \frac{2}{\eta},$$

allowing convergence to minima of any sharpness provided $\eta \leq 2$.

## 5. Evaluations

In this section, we demonstrate that SASSHA can indeed improve upon existing second-order methods available for standard deep learning tasks. We also show that SASSHA performs competitively to the first-order baseline methods. Specifically, SASSHA is compared to AdaHessian (Yao et al., 2021), Sophia-H (Liu et al., 2024), Shampoo (Gupta et al., 2018), SGD, AdamW (Loshchilov & Hutter, 2018), and SAM (Foret et al., 2021) on a diverse set of both vision and language tasks. We emphasize that we perform an *extensive* hyperparameter search to rigorously tune all optimizers and ensure fair comparisons. We provide the details of experiment settings to reproduce our results in Appendix E. The code to reproduce all results reported in this work is made available for download at `https://github.com/LOG-postech/Sassha`.

### 5.1. Image Classification

We first evaluate SASSHA for image classification on CIFAR-10, CIFAR-100, and ImageNet. We train various models of the ResNet family (He et al., 2016; Zagoruyko & Komodakis, 2016) and an efficient variant of Vision Transformer (Beyer et al., 2022). We adhere to standard inception-style data augmentations during training instead of making use of advanced data augmentation techniques (DeVries & Taylor, 2017) or regularization methods (Gastaldi, 2017). Results are presented in Table 2 and Figure 3.

We begin by comparing the generalization performance of adaptive second-order methods to that of first-order methods. Across all settings, adaptive second-order methods consistently exhibit lower accuracy than their first-order counterparts. This observation aligns with previous studies indicating that second-order optimization often result in poorer generalization compared to first-order approaches. In contrast, SASSHA, benefiting from sharpness minimization, consistently demonstrates superior generalization performance, outperforming both first-order and second-order methods in every setting. Particularly, SASSHA is up to 4% more effective than the best-performing adaptive or second-order methods (*e.g.*, WRN-28-10, ViT-s-32). In addition, SASSHA continually surpasses SGD and AdamW by approximately 0.3% to 3%, even when these methods are trained for twice as many epochs. Further details on these experiments are provided in Appendix H.

Interestingly, SASSHA also outperforms SAM. Since first-order methods typically exhibit superior generalization performance compared to second-order methods, it might be intuitive to expect SAM to surpass SASSHA if the two are viewed merely as the outcomes of applying sharpness minimization to first-order and second-order methods, respectively. However, the results conflict with this intuition. We attribute this to the careful design choices made in SASSHA, stabilizing Hessian approximation under sharpness minimization, so as to unleash the potential of the second-order method, leading to its outstanding performance. As a support, we show that naively incorporating SAM into other second-order methods does not yield these favorable results in Appendix I. We also make more comparisons with SAM in Section 5.3.

### 5.2. Language Modeling

Recent studies have shown the potential of second-order methods for pretraining language models. Here, we first evaluate how SASSHA performs on this task. Specifically, we train GPT1-mini, a scaled-down variant of GPT1 (Radford et al., 2019), on Wikitext-2 dataset (Merity et al., 2022) using various methods including SASSHA and compare their results (see the left of Table 3). Our results show that SASSHA achieves the lowest perplexity among all methods including Sophia-H (Liu et al., 2024), a recent method that is designed specifically for language modeling tasks and sets state of the art, which highlights generality in addition to the numerical advantage of SASSHA. We further evaluate SASSHA for GPT2 and provide results in Appendix J.

We also extend our evaluation to finetuning tasks. Specifically, we finetune SqueezeBERT (Iandola et al., 2020) for diverse tasks in the GLUE benchmark (Wang et al., 2018). The results are on the right side of Table 3. It shows that SASSHA compares competitively to other second-order methods. No-

*Table 2.* Image classification results of various optimization methods in terms of final validation accuracy (mean±std). SASSHA consistently outperforms the other methods for all workloads. * means *omitted* due to excessive computational requirements.

| Category | Method | CIFAR-10 | | CIFAR-100 | | ImageNet | |
| --- | --- | --- | --- | --- | --- | --- | --- |
| | | ResNet-20 | ResNet-32 | ResNet-32 | WRN-28-10 | ResNet-50 | ViT-s-32 |
| First-order | SGD | $92.03_{\pm0.32}$ | $92.69_{\pm0.06}$ | $69.32_{\pm0.19}$ | $80.06_{\pm0.15}$ | $75.58_{\pm0.05}$ | $62.90_{\pm0.36}$ |
| | AdamW | $92.04_{\pm0.11}$ | $92.42_{\pm0.13}$ | $68.78_{\pm0.22}$ | $79.09_{\pm0.35}$ | $75.38_{\pm0.08}$ | $66.46_{\pm0.15}$ |
| | SAM $_{SGD}$ | $92.85_{\pm0.07}$ | $93.89_{\pm0.13}$ | $71.99_{\pm0.20}$ | $83.14_{\pm0.13}$ | $76.36_{\pm0.16}$ | $64.54_{\pm0.63}$ |
| | SAM $_{AdamW}$ | $92.77_{\pm0.29}$ | $93.45_{\pm0.24}$ | $71.15_{\pm0.37}$ | $82.88_{\pm0.31}$ | $76.35_{\pm0.16}$ | $68.31_{\pm0.17}$ |
| Second-order | AdaHessian | $92.00_{\pm0.17}$ | $92.48_{\pm0.15}$ | $68.06_{\pm0.22}$ | $76.92_{\pm0.26}$ | $73.64_{\pm0.16}$ | $66.42_{\pm0.23}$ |
| | Sophia-H | $91.81_{\pm0.27}$ | $91.99_{\pm0.08}$ | $67.76_{\pm0.37}$ | $79.35_{\pm0.24}$ | $72.06_{\pm0.49}$ | $62.44_{\pm0.36}$ |
| | Shampoo | $88.55_{\pm0.83}$ | $90.23_{\pm0.24}$ | $64.08_{\pm0.46}$ | $74.06_{\pm1.28}$ | * | * |
| | SASSHA | $\mathbf{92.98}_{\pm0.05}$ | $\mathbf{94.09}_{\pm0.24}$ | $\mathbf{72.14}_{\pm0.16}$ | $\mathbf{83.54}_{\pm0.08}$ | $\mathbf{76.43}_{\pm0.18}$ | $\mathbf{69.20}_{\pm0.30}$ |

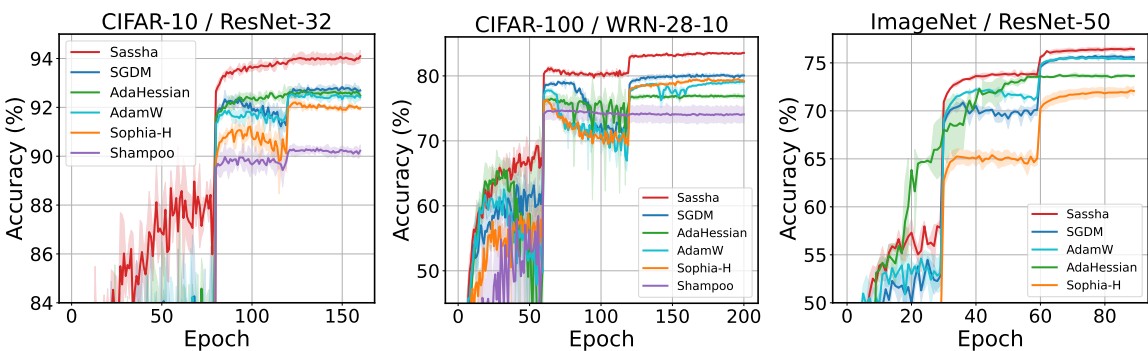

*Figure 3.* Validation accuracy curves along the training trajectory. We also provide loss curves in Appendix G.

*Table 3.* Language finetuning and pertraining results for various optimizers. For finetuning, SASSHA achieves better results than AdamW and AdaHessian and compares competitively with Sophia-H. For pretraining, SASSHA achieves the lowest perplexity among all optimizers.

| | Pretrain/ GPT1-mini | Finetune / SqeezeBERT | | | | | | |
| --- | --- | --- | --- | --- | --- | --- | --- | --- |
| | Wikitext-2 | SST-2 | MRPC | STS-B | QQP | MNLI | QNLI | RTE |
| | Perplexity | Acc | Acc / F1 | S/P corr. | F1 / Acc | mat/m.mat | Acc | Acc |
| AdamW | $175.06_{\pm0.19}$ | $90.29_{\pm0.52}$ | $84.56_{\pm0.25}$ / $88.99_{\pm0.11}$ | $88.34_{\pm0.15}$ / $88.48_{\pm0.20}$ | $89.92_{\pm0.05}$ / $86.58_{\pm0.11}$ | $81.22_{\pm0.07}$ / $82.26_{\pm0.05}$ | $89.93_{\pm0.14}$ | $68.95_{\pm0.72}$ |
| SAM $_{AdamW}$ | $158.06_{\pm0.23}$ | $\mathbf{90.52}_{\pm0.27}$ | $83.25_{\pm2.79}$ / $87.90_{\pm2.21}$ | $88.38_{\pm0.01}$ / $88.79_{\pm0.99}$ | $90.26_{\pm0.28}$ / $86.99_{\pm0.31}$ | $81.56_{\pm0.18}$ / $\mathbf{82.46}_{\pm0.19}$ | $\mathbf{90.38}_{\pm0.05}$ | $68.83_{\pm1.46}$ |
| AdaHessian | $407.69_{\pm0.20}$ | $89.64_{\pm0.13}$ | $79.74_{\pm4.00}$ / $85.26_{\pm3.50}$ | $86.08_{\pm4.04}$ / $86.46_{\pm4.06}$ | $90.37_{\pm0.05}$ / $87.07_{\pm0.05}$ | $81.33_{\pm0.17}$ / $82.08_{\pm0.02}$ | $89.94_{\pm0.12}$ | $71.00_{\pm1.04}$ |
| Sophia-H | $157.60_{\pm0.37}$ | $90.44_{\pm0.46}$ | $85.78_{\pm1.07}$ / $89.90_{\pm0.82}$ | $88.17_{\pm1.07}$ / $88.53_{\pm1.13}$ | $90.70_{\pm0.04}$ / $87.60_{\pm0.06}$ | $\mathbf{81.77}_{\pm0.18}$ / $82.36_{\pm0.22}$ | $90.12_{\pm0.14}$ | $70.76_{\pm1.44}$ |
| SASSHA | $\mathbf{122.40}_{\pm0.16}$ | $90.44_{\pm0.98}$ | $\mathbf{86.28}_{\pm0.28}$ / $\mathbf{90.13}_{\pm0.161}$ | $\mathbf{88.72}_{\pm0.75}$ / $\mathbf{89.10}_{\pm0.70}$ | $\mathbf{90.91}_{\pm0.06}$ / $\mathbf{87.85}_{\pm0.09}$ | $81.61_{\pm0.25}$ / $81.71_{\pm0.11}$ | $89.85_{\pm0.20}$ | $\mathbf{72.08}_{\pm0.55}$ |

tably, it also outperforms AdamW—often the method of choice for training language models—on nearly all tasks.

## 5.3. Comparison to SAM

So far, we have seen that SASSHA outperforms second-order methods quite consistently on both vision and language tasks. Interestingly, we also find that SASSHA often improves upon SAM. In particular, it appears that the gain is larger for the Transformer-based architectures, *i.e.*, ViT results in Table 2 or GPT/BERT results in Table 3.

To further investigate these findings, we conducted additional experiments. First, we allocate more training budgets

to SAM to see whether it compares to SASSHA. The results are presented in Table 4. We find that SAM still underperforms SASSHA, even though it is given more budgets of training iterations over data or wall-clock time. Furthermore, we also compare SASSHA to more advanced variants of SAM including ASAM (Kwon et al., 2021) and GSAM (Zhuang et al., 2022), showing that SASSHA performs competitively even to these methods (Appendix K). Notably, however, these variants of SAM require a lot more hyperparameter tuning to be compared.

We suspect that this may be due to the robustness of SASSHA to the block heterogeneity inherent in Transformer archi-

*Table 4.* Comparison between SASSHA and SAM with more training budgets for the ViT-s-32 / ImageNet workload.

|  | Epoch | Time (s) | Accuracy (%) |
|---|---|---|---|
| SAM $_{\text{SGD}}$ | 180 | 220,852 | $65.403_{\pm 0.63}$ |
| SAM $_{\text{AdamW}}$ | 180 | 234,374 | $68.706_{\pm 0.16}$ |
| SASSHA | **90** | **123,948** | **$69.195_{\pm 0.30}$** |

tectures, where the Hessian spectrum varies significantly across different blocks. This characteristic is known to make SGD perform worse than adaptive methods like Adam on Transformer-based models (Zhang et al., 2024). Since SASSHA leverages second-order information via preconditioning gradients, it has the potential to address the ill-conditioned nature of Transformers more effectively than SAM with first-order methods.

# 6. Further Analysis

## 6.1. Robustness

Noisily labeled training data can critically degrade generalization performance (Natarajan et al., 2013). To evaluate how SASSHA generalizes under these practical conditions, we randomly corrupt certain fractions of the training data and compare the validation performances between different methods. The results show that SASSHA outperforms other methods across all noise levels with minimal accuracy degradation (Table 5). Additionally, we also observe the same trend on CIFAR-10 (Table 21).

Interestingly, SASSHA surpasses SAM (Foret et al., 2021), which is known to be one of the most robust techniques against label noise (Baek et al., 2024). We hypothesize that its robustness stems from the complementary benefits of the sharpness-minimization scheme and second-order methods. Specifically, SAM enhances robustness by adversarially perturbing the parameters and giving more importance to clean data during optimization, making the model more resistant to label noise (Foret et al., 2021; Baek et al., 2024). Also, recent research indicates that second-order methods are robust to label noise due to preconditioning that reduces the variance in the population risk (Amari et al., 2021).

## 6.2. Stability

To show the effect of the square-root function on stabilizing the training process, we run SASSHA without the square-root (No-Sqrt), repeatedly for multiple times with different random seeds. As a result, we find that the training diverges most of the time. A failure case is depicted in Figure 4.

At first, we find that the level of training loss for No-Sqrt is much higher than that of SASSHA, and also, it spikes up

*Table 5.* Validation accuracy measured for ResNet-32/CIFAR-100 at different levels of noise. SASSHA shows the best robustness.

| | Noise level | | | |
|---|---|---|---|---|
| Method | 0% | 20% | 40% | 60% |
| SGD | $69.32_{\pm 0.19}$ | $62.18_{\pm 0.06}$ | $55.78_{\pm 0.55}$ | $45.53_{\pm 0.78}$ |
| SAM $_{\text{SGD}}$ | $71.99_{\pm 0.20}$ | $65.53_{\pm 0.11}$ | $61.20_{\pm 0.17}$ | $51.93_{\pm 0.47}$ |
| AdaHessian | $68.06_{\pm 0.22}$ | $63.06_{\pm 0.25}$ | $58.37_{\pm 0.13}$ | $46.02_{\pm 1.96}$ |
| Sophia-H | $67.76_{\pm 0.37}$ | $62.34_{\pm 0.47}$ | $56.54_{\pm 0.28}$ | $45.37_{\pm 0.27}$ |
| Shampoo | $64.08_{\pm 0.46}$ | $58.85_{\pm 0.66}$ | $53.82_{\pm 0.71}$ | $42.91_{\pm 0.99}$ |
| SASSHA | **$72.14_{\pm 0.16}$** | **$66.78_{\pm 0.47}$** | **$61.97_{\pm 0.27}$** | **$53.98_{\pm 0.57}$** |

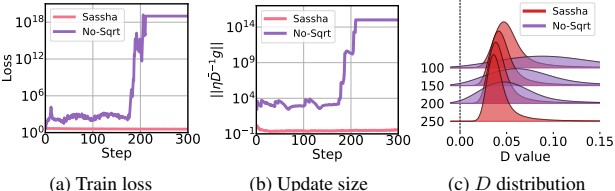

| (a) Train loss | (b) Update size | (c) $D$ distribution |
|---|---|---|

*Figure 4.* Effects of square-root measured for ResNet-32/CIFAR-100; $D$ is set to be either $|\widehat{H}|^{1/2}$ for SASSHA or $|\widehat{H}|$ for No-Sqrt. Sharpness minimization drives the diagonal Hessian entries move towards zero, causing divergence. The square-root in SASSHA helps counteract this effect, stabilizing the training process.

around step 200 (Figure 4a). To look into it further, we also measure the update sizes along the trajectory (Figure 4b). The results show that it matches well with the loss curves, suggesting that the training failure is somehow due to taking too large steps.

It turns out that this problem stems from the preconditioning matrix $D$ being too small; *i.e.*, the distribution of diagonal entries in the preconditioning matrix gradually shifts toward zero values (Figure 4c); as a result, $D^{-1}$ becomes too large, creating large steps. This progressive increase in near-zero diagonal Hessian entries is precisely due to the sharpness minimization scheme that we introduced; it penalizes the Hessian eigenspectrum to yield flat solutions, yet it could also make training unstable if taken naively. By including square-root, the preconditioner are less situated near zero, effectively suppressing the risk of large updates, thereby stabilizing the training process. We validate this further by showing its superiority to other alternatives including damping and clipping in Appendix F.1.

We also provide an ablation analysis for the absolute-value function in Appendix F.2, which demonstrates that it increases the stability of SASSHA in tandem with square-root.

## 6.3. Efficiency

Here we show the effectiveness of lazy Hessian updates in SASSHA. The results are shown in Figure 5. At first, we see that SASSHA maintains its performance even at $k = 100$,

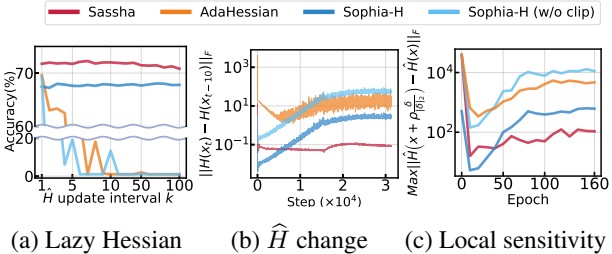

(a) Lazy Hessian    (b) $\widehat{H}$ change    (c) Local sensitivity

*Figure 5.* Effect of lazy Hessian for ResNet-32/CIFAR-100. SASSHA stays within the region where the Hessian varies small.

indicating that it is extremely robust to lazy Hessian updates (Figure 5a). We also measure the difference between the current and previous Hessians to validate lazy Hessian updates more directly (Figure 5b). The result shows that SASSHA keeps the changes in Hessian to be small, and much smaller than other methods, indicating its advantage of robust reuse, and hence, computational efficiency.

We attribute this robustness to the sharpness minimization scheme incorporated in SASSHA, which can potentially bias optimization toward the region of low curvature sensitivity. To verify, we define local Hessian sensitivity as follows:

$$\max_{\delta \sim \mathcal{N}(0,1)} \left\| \widehat{H}\left(x + \rho \frac{\delta}{\|\delta\|_2}\right) - \widehat{H}(x) \right\|_F \qquad (7)$$

*i.e.*, it measures the maximum change in Hessian induced from normalized random perturbations. A smaller Hessian sensitivity would suggest reduced variability in the loss curvature, leading to greater relevance of the current Hessian for subsequent optimization steps. We find that SASSHA is far less sensitive compared to other methods (Figure 5c).

### 6.4. Cost

Second-order methods can be highly costly. In this section, we discuss the computational cost of SASSHA and reveal its competitiveness to other methods.

SASSHA requires one gradient computation (GC) in the sharpness minimization step, one Hessian-vector product (HVP) for diagonal Hessian computation, and an additional GC in the descent step. That is, a total of 2GCs and 1HVP are required. However, with lazy Hessian updates, the number of HVPs reduces drastically to $1/k$. With $k = 10$ as the default value used in this work, this scales down to 0.1HVPs.

It turns out that this is critical to the utility of SASSHA, because 1HVP is known to take about $\times 3$ the computation time of 1GC in practice (Dagréou et al., 2024). Compared to conventional second-order methods (1GC + 1HVP ≃ 4GCs), the cost of SASSHA can roughly be a half of that (2.3GCs). It is also comparable to standard SAM variants (2GCs).

*Table 6.* Average wall-clock time per epoch (s) and the theoretical cost of different methods. SASSHA can be an effective alternative to existing methods for its enhanced generalization performance.

| Method | Cost | | | | CIFAR10 | CIFAR100 | ImageNet |
|---|---|---|---|---|---|---|---|
| | Descent | Sharpness | Hessian | Total | ResNet32 | WRN28-10 | ViT-small |
| AdamW | 1 GC | 0 GC | 0 HVP | 1 GC | 5.03 | 59.29 | 976.56 |
| SAM | 1 GC | 1 GC | 0 HVP | 2 GC | 9.16 | 118.46 | 1302.08 |
| AdaHessian | 1 GC | 0 GC | 1 HVP | 4 GC | 33.75 | 296.63 | 2489.07 |
| SASSHA | 1 GC | 1 GC | 0.1 HVP | 2.3 GC | 12.00 | 142.06 | 1377.20 |
| M-SASSHA | 1 GC | 0 GC | 0.1 HVP | 1.3 GC | 8.91 | 84.12 | 1065.40 |

Furthermore, we can leverage a momentum of gradients in the perturbation step to reduce the cost. This variant M-SASSHA requires only 1.3GCs with minimal decrease in performance. Notably, M-SASSHA still outperforms standard first-order methods like SGD and AdamW (Appendix M).

To verify, we measure the average wall-clock times and present the results in Table 6. First, one can see that the theoretical cost is reflected well on the actual cost; *i.e.*, the time measurements scales proportionally roughly well with respect to the total cost. More importantly, this result indicates the potential of SASSHA for performance-critical applications. Considering its well-balanced cost, and that it has been challenging to employ second-order methods efficiently for large-scale tasks without sacrificing performance, SASSHA can be a reasonable addition to the lineup.

## 7. Conclusion

In this work, we focus on addressing the issue of poor generalization in approximate second-order methods. Our empirical analysis indicates that this limitation may be attributed to their tendency to converge to sharp minima, which are known to correlate with weaker generalization performance. To this end, we propose a new method called SASSHA that stably minimizes sharpness within the framework of second-order optimization. SASSHA converges to flat solutions and achieves state-of-the-art performance within this class. SASSHA also performs competitively to widely-used first-order, adaptive, and sharpness-aware methods. SASSHA achieves this efficiently through lazy Hessian updates, to which it is robust, and does so without requiring extra hyperparameter tuning. Moreover, SASSHA exhibits strong resilience to label noise. All of these are rigorously assessed with extensive experiments.

Nonetheless, there are still many limitations to be addressed to further improve this work. Some examples may include, but are not limited to, extending experiments to various models and different data of extreme scales, as well as developing theoretical properties such as convergence rate, generalization bound, and implicit bias, all to more rigorously confirm the value of SASSHA. Seeing it as an exciting opportunity, we plan to investigate further in future work.

## Acknowledgement

This work was partly supported by the Institute of Information & communications Technology Planning & Evaluation (IITP) grant funded by the Korean government (MSIT) (IITP-2019-0-01906, Artificial Intelligence Graduate School Program (POSTECH), RS-2024-00338140, Development of learning and utilization technology to reflect sustainability of generative language models and up-to-dateness over time), and the National Research Foundation of Korea (NRF) grant funded by the Korean government (MSIT) ( NRF-2022R1F1A1064569, RS-2023-00210466, RS-2023-00265444).

## Impact statement

This paper contributes to advancing second-order optimization, with potential implications for both theoretical insights and practical applications. While our work does not present immediate concerns warranting specific emphasis, we recognize that progress in this field may have broader societal impacts. We remain committed to engaging in discussions on the broader implications of our research should the need arise in the future.

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

# A. Sharpness Measurements for Other Settings

*Table 7.* Sharpness measurements of the solutions found by seven different optimizers and their generalization on CIFAR-10/100 and Wikitext-2. Approximate second-order methods tend to yield highly sharp solutions and poor generalization compared to SGD; SASSHA and M-SASSHA effectively recover this. Here, we measure sharpness in terms of maximum Hessian eigenvalue $\lambda_{max}(H)$, trace of Hessian $\text{tr}(H)$, gradient-direction sharpness $\delta L_{\text{grad}}$, and average sharpness $\delta L_{\text{avg}}$, along with generalization using validation loss $L_{val}$ and accuracy $\text{Acc}_{val}$.

| | | Sharpness | | | | Generalization | |
|---|---|---|---|---|---|---|---|
| | | $\lambda_{max}(H)$ | $\text{tr}(H)_{\times 10^3}$ | $\delta L_{\text{grad}}$ | $\delta L_{\text{avg}\times 10^{-3}}$ | $L_{\text{val}}$ | $\text{Acc}_{\text{val}}$ |
| | | | | CIFAR-10 | | | |
| | SGD | $107_{\pm 4.370}$ | $1.380_{\pm 0.010}$ | $0.840_{\pm 0.304}$ | $0.690_{\pm 0.390}$ | $0.295_{\pm 0.008}$ | $92.03_{\pm 0.32}$ |
| | SAM | $58_{\pm 2.980}$ | $0.730_{\pm 0.040}$ | $0.171_{\pm 0.038}$ | $0.461_{\pm 0.240}$ | $0.119_{\pm 0.002}$ | $92.85_{\pm 0.07}$ |
| | AdaHessian | $23048_{\pm 29932}$ | $189.5_{\pm 240.6}$ | $4.538_{\pm 1.634}$ | $198.7_{\pm 266.0}$ | $0.260_{\pm 0.006}$ | $92.00_{\pm 0.17}$ |
| ResNet20 | Sophia-H | $3606_{\pm 303.0}$ | $31.24_{\pm 2.628}$ | $6.120_{\pm 1.634}$ | $18.11_{\pm 1.000}$ | $0.316_{\pm 0.002}$ | $91.81_{\pm 0.27}$ |
| | Shampoo | $647066_{\pm 419964}$ | $3900_{\pm 1825}$ | $166.3_{\pm 48.00}$ | $2177189_{\pm 1628993}$ | $0.381_{\pm 0.028}$ | $88.55_{\pm 0.83}$ |
| | M-SASSHA | $129_{\pm 17.00}$ | $1.580_{\pm 0.080}$ | $1.551_{\pm 0.684}$ | $1.025_{\pm 0.360}$ | $0.234_{\pm 0.003}$ | $92.36_{\pm 0.23}$ |
| | SASSHA | $78_{\pm 5.090}$ | $0.860_{\pm 0.030}$ | $0.184_{\pm 0.053}$ | $0.388_{\pm 0.704}$ | $0.209_{\pm 0.001}$ | $92.98_{\pm 0.05}$ |
| | SGD | $56_{\pm 5.100}$ | $0.800_{\pm 0.040}$ | $0.560_{\pm 0.219}$ | $0.196_{\pm 0.146}$ | $0.309_{\pm 0.002}$ | $92.69_{\pm 0.06}$ |
| | SAM | $45_{\pm 2.670}$ | $0.580_{\pm 0.020}$ | $0.107_{\pm 0.005}$ | $0.753_{\pm 0.351}$ | $0.128_{\pm 0.001}$ | $93.89_{\pm 0.13}$ |
| | AdaHessian | $1746_{\pm 1018}$ | $17.06_{\pm 10.24}$ | $4.599_{\pm 1.710}$ | $5.518_{\pm 3.623}$ | $0.278_{\pm 0.006}$ | $92.48_{\pm 0.15}$ |
| ResNet32 | Sophia-H | $7167_{\pm 2755}$ | $18.82_{\pm 5.500}$ | $9.399_{\pm 2.283}$ | $7.915_{\pm 3.397}$ | $0.394_{\pm 0.010}$ | $91.99_{\pm 0.08}$ |
| | Shampoo | $717553_{\pm 93129}$ | $4523_{\pm 629.7}$ | $162.1_{\pm 123.2}$ | $105322_{\pm 82246}$ | $0.348_{\pm 0.008}$ | $90.23_{\pm 0.24}$ |
| | M-SASSHA | $283_{\pm 10.00}$ | $3.960_{\pm 0.100}$ | $2.986_{\pm 1.133}$ | $1.300_{\pm 0.969}$ | $0.211_{\pm 0.010}$ | $93.18_{\pm 0.30}$ |
| | SASSHA | $47_{\pm 1.880}$ | $0.590_{\pm 0.020}$ | $0.136_{\pm 0.019}$ | $0.714_{\pm 0.090}$ | $0.177_{\pm 0.002}$ | $94.09_{\pm 0.24}$ |
| | | | | CIFAR-100 | | | |
| | SGD | $265_{\pm 25.00}$ | $7.290_{\pm 0.300}$ | $0.703_{\pm 0.132}$ | $1.310_{\pm 1.030}$ | $1.260_{\pm 0.001}$ | $69.32_{\pm 0.19}$ |
| | SAM | $123_{\pm 11.00}$ | $2.630_{\pm 0.090}$ | $0.266_{\pm 0.025}$ | $-0.619_{\pm 0.594}$ | $0.512_{\pm 0.016}$ | $71.99_{\pm 0.20}$ |
| | AdaHessian | $11992_{\pm 5779}$ | $46.94_{\pm 17.60}$ | $4.119_{\pm 1.136}$ | $12.50_{\pm 6.080}$ | $1.377_{\pm 0.070}$ | $68.06_{\pm 0.22}$ |
| ResNet32 | Sophia-H | $22797_{\pm 10857}$ | $68.15_{\pm 20.19}$ | $8.130_{\pm 3.082}$ | $19.19_{\pm 6.380}$ | $1.463_{\pm 0.022}$ | $67.76_{\pm 0.37}$ |
| | Shampoo | $436374_{\pm 9017}$ | $6823_{\pm 664.7}$ | $73.27_{\pm 12.51}$ | $49307489_{\pm 56979794}$ | $1.386_{\pm 0.010}$ | $64.08_{\pm 0.46}$ |
| | M-SASSHA | $382_{\pm 65.00}$ | $8.750_{\pm 0.310}$ | $2.391_{\pm 0.425}$ | $2.260_{\pm 1.660}$ | $1.067_{\pm 0.001}$ | $70.93_{\pm 0.21}$ |
| | SASSHA | $107_{\pm 40.00}$ | $1.870_{\pm 0.650}$ | $0.238_{\pm 0.088}$ | $0.650_{\pm 0.860}$ | $0.961_{\pm 0.005}$ | $72.14_{\pm 0.16}$ |
| | SGD | $18_{\pm 1.170}$ | $0.660_{\pm 0.040}$ | $1.984_{\pm 0.506}$ | $-0.007_{\pm 0.028}$ | $0.820_{\pm 0.005}$ | $80.06_{\pm 0.15}$ |
| | SAM | $9_{\pm 0.866}$ | $0.230_{\pm 0.010}$ | $0.841_{\pm 0.084}$ | $0.024_{\pm 0.041}$ | $0.648_{\pm 0.006}$ | $83.14_{\pm 0.13}$ |
| | AdaHessian | $35119_{\pm 46936}$ | $139.5_{\pm 191.0}$ | $6.745_{\pm 1.932}$ | $19.727_{\pm 27.87}$ | $1.005_{\pm 0.008}$ | $76.92_{\pm 0.26}$ |
| WRN28-10 | Sophia-H | $3419_{\pm 3240}$ | $13.57_{\pm 3.300}$ | $5.073_{\pm 0.268}$ | $0.067_{\pm 0.054}$ | $0.866_{\pm 0.003}$ | $79.35_{\pm 0.24}$ |
| | Shampoo | $102129_{\pm 60722}$ | $1459_{\pm 709.4}$ | $483.0_{\pm 172.0}$ | $98.558_{\pm 123.1}$ | $1.173_{\pm 0.088}$ | $74.06_{\pm 1.28}$ |
| | M-SASSHA | $2257_{\pm 248.0}$ | $30.40_{\pm 4.780}$ | $4.599_{\pm 0.003}$ | $0.301_{\pm 0.047}$ | $0.757_{\pm 0.011}$ | $81.53_{\pm 0.27}$ |
| | SASSHA | $84_{\pm 3.150}$ | $2.030_{\pm 0.110}$ | $4.540_{\pm 0.122}$ | $0.007_{\pm 0.129}$ | $0.625_{\pm 0.002}$ | $83.54_{\pm 0.08}$ |
| | | | | Wikitext-2 | | | |
| | AdamW | $836_{\pm 13.00}$ | $31.61_{\pm 0.433}$ | $1.642_{\pm 1.036}$ | $7_{\pm 0}$ | $5.072_{\pm 0.013}$ | $175.06_{\pm 0.19}$ |
| | AdaHessian | $13141_{\pm 14432}$ | $46.36_{\pm 26.85}$ | $0.289_{\pm 0.187}$ | $9_{\pm 5}$ | $7.231_{\pm 0.043}$ | $407.69_{\pm 0.20}$ |
| Mini-GPT1 | Sophia-H | $319_{\pm 14.00}$ | $55.17_{\pm 1.100}$ | $0.824_{\pm 0.089}$ | $13_{\pm 1}$ | $5.077_{\pm 0.014}$ | $157.60_{\pm 0.37}$ |
| | M-SASSHA | $145_{\pm 125.0}$ | $13.23_{\pm 17.19}$ | $0.379_{\pm 0.275}$ | $3_{\pm 1}$ | $5.259_{\pm 0.010}$ | $125.01_{\pm 0.21}$ |
| | SASSHA | $79_{\pm 2.000}$ | $14.50_{\pm 0.325}$ | $0.221_{\pm 0.023}$ | $3_{\pm 0}$ | $4.808_{\pm 0.001}$ | $122.40_{\pm 0.16}$ |

# B. Algorithm Comparison

*Table 8.* Comparison of various optimization algorithms in terms of gradient momentum $m_t$, diagonal preconditioning matrix $D_t$, and method-specific operations $\mathbf{U}(z)$. Here $g_t, \widehat{H}_t$ are the stochastic gradient and the Hessian estimation respectively, and $\beta_1, \beta_2$ denotes the momentum hyperparameters for the gradient and estimated Hessian. Bias correction $\mathtt{bc}(\cdot)$ compensates for initialization biases in the gradient and Hessian momentum variables due to zero initialization.

| | | $x_{t+1} = x_t - \eta_t \mathbf{U}(D_t^{-1} m_t)$ | |
|---|---|---|---|
| | $m_t$ | $D_t$ | $\mathbf{U}(z)$ |
| SGD with momentum | $\beta_1 m_{t-1} + (1-\beta_1)g_t$ | $I$ | $z$ |
| Stochastic Newton | $g_t$ | $H_t(x_t)$ | $z$ |
| Adam (Kingma & Ba, 2015) | $\beta_1 m_{t-1} + (1-\beta_1)g_t$ | $\sqrt{\beta_2 v_{t-1} + (1-\beta_2)\,\mathrm{diag}(g_t g_t^\top)}$ | $\mathtt{bc}(z)$ |
| AdaHessian (Yao et al., 2021) | " | $\sqrt{\beta_2 v_{t-1} + (1-\beta_2)\,\widehat{H}_t^{(s)}(x_t)^2}$ | $\mathtt{bc}(z)$ |
| Sophia-H (Liu et al., 2024) | " | $\beta_2 v_{t-1} + (1-\beta_2)\,\widehat{H}_t^{(c)}(x_t)$  every $k$ steps | $\mathrm{clip}(z)$ |
| SASSHA (Ours) | $\beta_1 m_{t-1} + (1-\beta_1)\,g_t(x_t + \boldsymbol{\epsilon_t^\star})$ | $\sqrt{\beta_2 v_{t-1} + (1-\beta_2)\,\lvert\widehat{H}_t(x_t + \boldsymbol{\epsilon_t^\star})\rvert}$  every $k$ steps | $\mathtt{bc}(z)$ |

In this section, we compare our algorithm with other adaptive and approximate second-order methods designed for deep learning to better illustrate our contributions within concurrent literature. We present a detailed comparison of each methods in Table 8.

Adam (Kingma & Ba, 2015) is an adaptive method popular among practitioners, which rescales the learning rate for each parameter dimension by dividing by the square root of the moving average of squared gradients. This adaptive learning rate effectively adjusts the gradient (momentum) at each descent step, accelerating convergence and improving update stability. Although Adam is not explicitly a second-order method, its process is related to second-order methods as it can be viewed as preconditioning via a diagonal approximation of the empirical Fisher information matrix. AdamW (Loshchilov & Hutter, 2018) proposes to improve Adam by decoupling the weight decay from the update rule for better generalization. This is also shown to be effective in most approximate second-order methods, thus employed in all subsequently mentioned algorithms.

AdaHessian (Yao et al., 2021) is one of the earliest approximate second-order optimization methods tailored for deep learning. To reduce the prohibitive cost of computing the Hessian, it uses Hutchinson's method (Hutchinson, 1989; Roosta-Khorasani & Ascher, 2014) to estimate a diagonal Hessian approximation $\widehat{H}_t$ and applies a moving average to reduce variance in the estimation. The authors also propose spatial averaging of the Hessian estimate, denoted as $(\widehat{H}_t^{(s)})$, which involves averaging the diagonal element within a filter of a convolution layer for filter-wise gradient scaling. Sophia (Liu et al., 2024) is an approximate second-order method specifically designed for language model pretraining. Its primary feature is the use of the clipping mechanism $\mathrm{clip}(z) = \max\{\min\{z,\rho\}, -\rho\}$ with a predefined threshold $\rho$ to control the worst-case update size resulting from errorneous diagonal Hessian estimates in preconditioning. Additionally, a hard adjustment is applied to each Hessian entry, substituting negative and very small values with a constant $\epsilon$, such as $\widehat{H}_t^{(c)} = \max\{\widehat{h}_t, \epsilon\}$ to prevent convergence to saddle points and mitigate numerical instability. Furthermore, Sophia also incorporates lazy Hessian updates to enhance computational efficiency. This works without significant performance degradation as the clipping technique and hard adjustment prevent a rapid change of the Hessian, keeping the previous Hessian relevant over more extended steps.

Our method, SASSHA, adds a perturbation $\epsilon_t^\star$ before computing the gradient and diagonal Hessian estimation to penalize sharpness during training for promoting improved generalization – an approach that, to our knowledge, has not been previously explored in the literature. However, naive second-order optimization under sharpness minimization can be numerically unstable, because, when the overall curvature is small due to sharpness reduction, Hessian underestimation can cause the optimization to yield extremely large update. To address this issue, we introduce two simple techniques to Hessian estimates: the square root and the absolute function. The square root smoothly adjusts underestimated curvature while preserving the relative scale among Hessian entries. The absolute function enforces Hessian estimates to be semi-positive

definite while maintaining their magnitude. This not only prevents convergence to saddle points or local maxima, but also allows the square root to operate on Hessian entries retaining the original Hessian magnitude. Together, the combination of sharpness minimization and Hessian stabilization enables efficient reuse of previously computed Hessians, resulting in a stable and efficient algorithm.

## C. Convergence Analysis of SASSHA

In this section, we provide preliminary convergence analysis results. Based on the well-established analyses of Li et al. (2023); Khanh et al. (2024), we further investigate the complexities arising from preconditioned perturbed gradients.

**Assumption C.1.** The function $f : \mathbb{R}^d \to \mathbb{R}$ is convex, $\beta$-smooth, and bounded from below, i.e., $f^* := \inf_x f(x) > -\infty$. Additionally, the gradient $\nabla f(x_t)$ is non-zero for a finite number of iterations, i.e., $\nabla f(x_t) \neq 0$ for all $t \in \{1, 2, \ldots, n\}$.

**Assumption C.2.** Step sizes $\eta_t$ and perturbation radii $\rho_t$ are assumed to satisfy the following conditions:

$$\sum_{t=1}^{\infty} \eta_t = \infty, \quad \sum_{t=1}^{\infty} \eta_t^2 < \infty, \quad \sum_{t=1}^{\infty} \rho_t^2 \eta_t < \infty.$$

*Remark* C.3. The following notations will be used throughout

1. $g_t := \nabla f(x_t)$ denotes the gradient of $f$ at iteration $t$.

2. The intermediate points and the difference between the gradients are defined as

$$x_{t+\frac{1}{2}} := x_t + \rho_t \frac{g_t}{\|g_t\|}, \quad g_{t+\frac{1}{2}} := \nabla f(x_{t+\frac{1}{2}}), \quad \delta_t := g_{t+\frac{1}{2}} - g_t.$$

3. For $u, v \in \mathbb{R}^d$, operations such as $\sqrt{v}$, $|v|$ and $\frac{v}{u}$, as well as the symbols $\preceq$ and $\succeq$, are applied element-wise.

*Remark* C.4. The update rule for the iterates is given by

$$x_{t+1} = x_t - \frac{\eta_t}{\sqrt{|\operatorname{diag}(\nabla^2 f(x_{t+\frac{1}{2}}))| + \epsilon}} \odot g_{t+\frac{1}{2}}, \tag{8}$$

where $\operatorname{diag}$ extracts the diagonal elements of a matrix as a vector, or constructs a diagonal matrix from a vector, and $\epsilon$ is a small constant introduced to prevent division by zero. Define $h_t$ as

$$h_t = \frac{\eta_t}{\sqrt{|\operatorname{diag}(\nabla^2 f(x_{t+\frac{1}{2}}))| + \epsilon}},$$

then the following hold

1. From the convexity and $\beta$-smoothness of $f$, the diagonal elements of $\nabla^2 f(x)$ are bounded within the interval $[0, \beta]$, i.e.,

$$0 \leq \left[\nabla^2 f(x)\right]_{(i,i)} = e_i^\top \nabla^2 f(x) e_i \leq \beta,$$

where $e_i$ is the $i$-th standard basis vector in $\mathbb{R}^d$.

2. The term $h_t$ is bounded as

$$\frac{\eta_t}{\sqrt{\beta} + \epsilon} \preceq h_t \preceq \frac{\eta_t}{\epsilon}.$$

*Remark* C.5. For the matrix representation

1. Denoting $H_t := \text{diag}(h_t)$, the matrix bounds for $H_t$ are given by

$$\frac{\eta_t}{\sqrt{\beta} + \epsilon} I \preceq H_t \preceq \frac{\eta_t}{\epsilon} I, \tag{9}$$

   where $I$ is the identity matrix.

2. Using the matrix notation $H_t$, the update for the iterates is expressed as

$$x_{t+1} = x_t - H_t g_{t + \frac{1}{2}}.$$

*Remark* C.6. From the $\beta$-smoothness of $f$, $\delta_t$ is bounded by

$$\|\delta_t\| \leq \beta \|x_t + \rho_t \frac{\nabla f(x_t)}{\|\nabla f(x_t)\|} - x_t\| = \beta \rho_t. \tag{10}$$

**Lemma C.7** (Descent Lemma). *Under Assumption C.1 and Assumption C.2, for given $\beta$ and $\epsilon$, there exists a $T \in \mathbb{N}$ such that for $\forall t \geq T$, $\eta_t$ satisfies $\eta_t \leq \min\left\{\frac{\epsilon^2}{6\beta(\sqrt{\beta}+\epsilon)}, \frac{\epsilon}{4\beta}\right\}$. For such $t \geq T$, the following inequality holds*

$$f(x_{t+1}) \leq f(x_t) - \frac{\eta_t}{2(\sqrt{\beta} + \epsilon)} \|g_t\|^2 + \frac{\eta_t}{\epsilon} \|\delta_t\|^2. \tag{11}$$

*Proof.* We begin by applying the $\beta$-smoothness of $f$,

$$\begin{aligned}
f(x_{t+1}) &\leq f(x_t) + \langle g_t, x_{t+1} - x_t \rangle + \frac{\beta}{2} \|x_{t+1} - x_t\|^2 \\
&= f(x_t) - \langle g_t, H_t(g_t + \delta_t) \rangle + \frac{\beta}{2} \|H_t(g_t + \delta_t)\|^2 \\
&\leq f(x_t) - g_t^\top H_t g_t + \frac{1}{2\alpha} g_t^\top H_t g_t + \frac{\alpha}{2} \delta_t^\top H_t \delta_t + \frac{\beta}{2} \|H_t(g_t + \delta_t)\|^2 \\
&\leq f(x_t) - (1 - \frac{1}{2\alpha}) \frac{\eta_t}{\sqrt{\beta} + \epsilon} \|g_t\|^2 + \frac{\alpha}{2} \frac{\eta_t}{\epsilon} \|\delta_t\|^2 + \frac{\beta}{2} \frac{\eta_t^2}{\epsilon^2} \|g_t + \delta_t\|^2 \\
&\leq f(x_t) - (1 - \frac{1}{2\alpha}) \frac{\eta_t}{\sqrt{\beta} + \epsilon} \|g_t\|^2 + \frac{\alpha}{2} \frac{\eta_t}{\epsilon} \|\delta_t\|^2 + \beta \frac{\eta_t^2}{\epsilon^2} (\|g_t\|^2 + \|\delta_t\|^2) \\
&= f(x_t) - \eta_t ((1 - \frac{1}{2\alpha}) \frac{1}{\sqrt{\beta} + \epsilon} - \beta \frac{\eta_t}{\epsilon^2}) \|g_t\|^2 + \eta_t (\frac{\alpha}{2\epsilon} + \beta \frac{\eta_t}{\epsilon^2}) \|\delta_t\|^2.
\end{aligned}$$

The second inequality follows from Young's inequality, the third inequality is obtained from Equation (9), and the last inequality is simplified using the property $\|a + b\|^2 \leq 2\|a\|^2 + 2\|b\|^2$. By setting $\alpha = \frac{3}{2}$, we get

$$= f(x_t) - \eta_t (\frac{2}{3} \left(\frac{1}{\sqrt{\beta} + \epsilon}\right) - \beta \frac{\eta_t}{\epsilon^2}) \|g_t\|^2 + \eta_t (\frac{3}{4\epsilon} + \beta \frac{\eta_t}{\epsilon^2}) \|\delta_t\|^2.$$

Since $\eta_t \to 0$, $\exists T \in \mathbb{N}$ such that $\eta_t \leq \min\{\frac{\epsilon^2}{6\beta(\sqrt{\beta}+\epsilon)}, \frac{\epsilon}{4\beta}\}$, this gives $\frac{2}{3}\left(\frac{1}{\sqrt{\beta}+\epsilon}\right) - \beta\frac{\eta_t}{\epsilon^2} \geq \frac{1}{2(\sqrt{\beta}+\epsilon)}$ and $\frac{3}{4\epsilon} + \beta\frac{\eta_t}{\epsilon^2} \leq \frac{1}{\epsilon}$, which implies

$$\leq f(x_t) - \frac{\eta_t}{2(\sqrt{\beta} + \epsilon)} \|g_t\|^2 + \frac{\eta_t}{\epsilon} \|\delta_t\|^2$$

$\square$

**Theorem C.8.** *Under Assumption C.1 and Assumption C.2, given any initial point $x_0 \in \mathbb{R}^d$, let $\{x_t\}$ be generated by Equation (8). Then, it holds that $\liminf_{t\to\infty} \|g_t\| = 0$.*

*Proof.* From Lemma C.7 and Equation (10), we have the bound

$$f(x_{t+1}) \leq f(x_t) - \frac{\eta_t}{2(\sqrt{\beta} + \epsilon)} \|g_t\|^2 + \frac{\eta_t}{\epsilon} \|\delta_t\|^2$$

$$\leq f(x_t) - \frac{\eta_t}{2(\sqrt{\beta} + \epsilon)} \|g_t\|^2 + \frac{\eta_t}{\epsilon} \beta^2 \rho_t^2.$$

By rearranging the terms, we obtain the following

$$\frac{\eta_t}{2(\sqrt{\beta} + \epsilon)} \|g_t\|^2 \leq f(x_t) - f(x_{t+1}) + \frac{\eta_t}{\epsilon} \beta^2 \rho_t^2.$$

For any $M > T$, we have

$$\frac{1}{2(\sqrt{\beta} + \epsilon)} \sum_{t=T}^{M} \eta_t \|g_t\|^2 \leq \sum_{t=T}^{M} (f(x_t) - f(x_{t+1})) + \frac{\beta^2}{\epsilon} \sum_{t=T}^{M} \rho_t^2 \eta_t$$

$$= f(x_T) - f(x_{M+1}) + \frac{\beta^2}{\epsilon} \sum_{t=T}^{M} \rho_t^2 \eta_t$$

$$\leq f(x_T) - \inf_{t \in \mathbb{N}} f(x_t) + \frac{\beta^2}{\epsilon} \sum_{t=T}^{M} \rho_t^2 \eta_t.$$

As $M \to \infty$, the series $\sum_{t=T}^{\infty} \eta_t \|g_t\|^2$ converges. Now, assume for contradiction that $\liminf_{t\to\infty} \|g_t\| \neq 0$. This means there exists some $\xi > 0$ and $N \geq T$ such that $\|g_t\| \geq \xi$ for all $t \geq N$. Consequently, we have

$$\infty > \sum_{t=N}^{\infty} \eta_t \|g_t\|^2 \geq \xi^2 \sum_{t=N}^{\infty} \eta_t = \infty,$$

which is a contradiction. Therefore, $\liminf_{t\to\infty} \|g_t\| = 0$. $\qquad\square$

# D. Linear stability analysis on SASSHA

In this section, we provide the detailed proof of Section 4.6.

We begin by considering the minimization of the training error

$$f(x) = \frac{1}{n}\sum_{i=1}^{n} f_i(x)$$

via a general second-order optimization method:

$$x_{t+1} = x_t - P(x_t; \xi_t)^{-1}\nabla f(x_t; \xi_t) \tag{12}$$

where $\xi_t$ is a *i.i.d.* random variable independent of $x_t$. For SASSHA, the update is specified as

$$x_{t+1} = x_t - \eta \frac{1}{\sqrt{|\operatorname{diag}(\nabla^2 f_{\xi_t}(x_t + \rho \nabla f_{\xi_t}(x_t)))| + \epsilon}} \odot \nabla f_{\xi_t}(x_t + \rho \nabla f_{\xi_t}(x_t)), \tag{13}$$

where $\operatorname{diag}$ extracts the diagonal elements of a matrix as a vector, or constructs a diagonal matrix from a vector, and $\epsilon$ is a small constant introduced to prevent division by zero.

We now derive the linear stability conditions for the fixed points of SASSHA.

**Definition D.1.** (Fixed point). A point $x^\star$ is called a fixed point of the stochastic dynamics (12) if, for any $\xi$, we have $\nabla f_\xi(x^\star) = 0$.

**Definition D.2.** (Linearized SASSHA). Let $x^\star$ be the fixed point of interest and assume $f(x^\star) = 0$. Consider the quadratic approximation of $f$ near $x^\star$: $f_\xi(x) \approx \frac{1}{2}(x - x^\star)^\top H_\xi(x - x^\star)$ with $H_\xi = \nabla^2 f_{\xi_t}(x^\star)$. The corresponding linearized SASSHA is given by

$$x_{t+1} = x_t - \eta \frac{1}{\sqrt{\operatorname{diag}(H_{\xi_t})} + \epsilon} \odot H_{\xi_t}(\tilde{x}_t - x^\star) \tag{14}$$

where $\tilde{x}_t = x_t + \rho H_{\xi_t}(x_t - x^\star)$ is the linearized perturbed point.

For brevity, we define

$$d_{\xi_t} := \sqrt{\operatorname{diag}(H_{\xi_t})} + \epsilon,$$

Then, the update (14) can be rewritten as

$$x_{t+1} = x_t - \eta \frac{1}{d_{\xi_t}} \odot H_{\xi_t}(\tilde{x}_t - x^\star), \tag{15}$$

*Remark* D.3. In a neighborhood of $x^\star$, the inverse scaling vector is uniformly upper bounded by

$$\frac{1}{d_{\xi_t}} \preceq \frac{1}{\epsilon}\mathbf{1} \tag{16}$$

where $\mathbf{1}$ is the vector of all ones.

**Definition D.4.** (Linear stability). Consider a fixed point $x^\star$ of linearized stochastic dynamic such as (14). We say that $x^\star$ is *linearly stable* if there exists a constant $C$ such that

$$\mathbb{E}\big[\|x_t - x^\star\|^2\big] \leq C\|x_0 - x^\star\|^2, \text{ for all } t > 0$$

**Theorem D.5.** *Assume without loss of generality that $x^\star = 0$. Then, the fixed point $x^\star$ of SASSHA is linearly stable if:*

$$\lambda_{max}\left(\left(I - \frac{\eta}{\epsilon}H - \frac{\eta\rho}{\epsilon}H^2\right)^2 + \frac{\eta^2 - 2\eta\rho\epsilon}{\epsilon^2}\left(\mathbb{E}H_{\xi_t}^2 - H^2\right) + \frac{2\eta^2\rho}{\epsilon^2}\left(\mathbb{E}H_{\xi_t}^3 - H^3\right) + \frac{\eta^2\rho^2}{\epsilon^2}\left(\mathbb{E}H_{\xi_t}^4 - H^4\right)\right) \leq 1.$$

*Proof.* Our goal is to obtain a bound of the form $\mathbb{E}\|x_t\|^2 \leq C\|x_0\|^2$, for some constant $C$. We begin by substituting (15) into $\mathbb{E}\|x_{t+1}\|^2$ and proceed to expand the terms as follows:

$$
\begin{aligned}
\mathbb{E}\left[\|x_{t+1}\|^2 \Big| x_t\right] &= x_t^\top \mathbb{E}\left[\left(I - \eta \operatorname{diag}(\frac{1}{d_{\xi_t}})H_{\xi_t}(I + \rho\,H_{\xi_t})\right)^\top \left(I - \eta \operatorname{diag}(\frac{1}{d_{\xi_t}})H_{\xi_t}(I + \rho\,H_{\xi_t})\right)|x_t\right] x_t \\
&\overset{(16)}{\leq} x_t^\top \mathbb{E}\left[\left(I - \eta\frac{1}{\epsilon}H_{\xi_t}(I + \rho\,H_{\xi_t})\right)^\top \left(I - \eta\frac{1}{\epsilon}H_{\xi_t}(I + \rho\,H_{\xi_t})\right)|x_t\right] x_t \\
&= x_t^\top \mathbb{E}\left[I - \frac{2\eta}{\epsilon}H_{\xi_t}(I + \rho\,H_{\xi_t}) + \frac{\eta^2}{\epsilon^2}H_{\xi_t}^2(I + \rho\,H_{\xi_t})^2|x_t\right] x_t \\
&= x_t^\top \mathbb{E}\left[I - \frac{2\eta}{\epsilon}H_{\xi_t} - \frac{2\eta\rho}{\epsilon}H_{\xi_t}^2 + \frac{\eta^2}{\epsilon^2}H_{\xi_t}^2 + \frac{2\eta^2\rho}{\epsilon^2}H_{\xi_t}^3 + \frac{\eta^2\rho^2}{\epsilon^2}H_{\xi_t}^4|x_t\right] x_t \\
&= x_t^\top \Big(I - \frac{2\eta}{\epsilon}H + \frac{\eta^2 - 2\eta\rho\epsilon}{\epsilon^2}H^2 + \frac{2\eta^2\rho}{\epsilon^2}H^3 + \frac{\eta^2\rho^2}{\epsilon^2}H^4 \\
&\quad + \frac{\eta^2 - 2\eta\rho\epsilon}{\epsilon^2}\left(\mathbb{E}H_{\xi_t}^2 - H^2\right) + \frac{2\eta^2\rho}{\epsilon^2}\left(\mathbb{E}H_{\xi_t}^3 - H^3\right) + \frac{\eta^2\rho^2}{\epsilon^2}\left(\mathbb{E}H_{\xi_t}^4 - H^4\right)\Big) x_t \\
&= x_t^\top \Big(\left(I - \frac{\eta}{\epsilon}H - \frac{\eta\rho}{\epsilon}H^2\right)^2 + \frac{\eta^2 - 2\eta\rho\epsilon}{\epsilon^2}\left(\mathbb{E}H_{\xi_t}^2 - H^2\right) + \frac{2\eta^2\rho}{\epsilon^2}\left(\mathbb{E}H_{\xi_t}^3 - H^3\right) + \frac{\eta^2\rho^2}{\epsilon^2}\left(\mathbb{E}H_{\xi_t}^4 - H^4\right)\Big) x_t
\end{aligned}
$$

Since for any $x$ and any matrix $A$ the inequality $x^\top A x \leq \lambda_{max}(A)\|x\|^2$ with $\lambda_{max}(A)$ denoting the maximum eigenvalue of $A$, holds true, applying this inequality and taking the total expectation yields the following:

$$
\mathbb{E}\|x_{t+1}\|^2 \leq \lambda_{max}\Big(\left(I - \frac{\eta}{\epsilon}H - \frac{\eta\rho}{\epsilon}H^2\right)^2 + \frac{\eta^2 - 2\eta\rho\epsilon}{\epsilon^2}\left(\mathbb{E}H_{\xi_t}^2 - H^2\right) + \frac{2\eta^2\rho}{\epsilon^2}\left(\mathbb{E}H_{\xi_t}^3 - H^3\right) + \frac{\eta^2\rho^2}{\epsilon^2}\left(\mathbb{E}H_{\xi_t}^4 - H^4\right)\Big)\mathbb{E}\|x_t\|^2
$$

Recursively applying this bound gives

$$
\mathbb{E}\|x_t\|^2 = \lambda_{max}\Big(\left(I - \frac{\eta}{\epsilon}H - \frac{\eta\rho}{\epsilon}H^2\right)^2 + \frac{\eta^2 - 2\eta\rho\epsilon}{\epsilon^2}\left(\mathbb{E}H_{\xi_t}^2 - H^2\right) + \frac{2\eta^2\rho}{\epsilon^2}\left(\mathbb{E}H_{\xi_t}^3 - H^3\right) + \frac{\eta^2\rho^2}{\epsilon^2}\left(\mathbb{E}H_{\xi_t}^4 - H^4\right)\Big)^t \mathbb{E}\|x_0\|^2
$$

Here, we can see that $x^*$ is linearly stable if

$$
\lambda_{max}\Big(\left(I - \frac{\eta}{\epsilon}H - \frac{\eta\rho}{\epsilon}H^2\right)^2 + \frac{\eta^2 - 2\eta\rho\epsilon}{\epsilon^2}\left(\mathbb{E}H_{\xi_t}^2 - H^2\right) + \frac{2\eta^2\rho}{\epsilon^2}\left(\mathbb{E}H_{\xi_t}^3 - H^3\right) + \frac{\eta^2\rho^2}{\epsilon^2}\left(\mathbb{E}H_{\xi_t}^4 - H^4\right)\Big) \leq 1.
$$

$\square$

# E. Experiment Setting

Here, we describe our experiment settings in detail. We evaluate SASSHA against AdaHessian (Yao et al., 2021), Sophia-H (Liu et al., 2024), Shampoo (Gupta et al., 2018), SGD, AdamW (Loshchilov & Hutter, 2018), and SAM (Foret et al., 2021) across a diverse set of vision and language tasks. Across all evaluations except for language finetuning, we set lazy Hessian update interval to $k = 10$ for SASSHA. In fact, Sophia-H also supports lazy Hessian updates, but Liu et al. (2024) reports that it achieves the best performance when $k = 1$, without lazy updating. Since our goal is to demonstrate that SASSHA exhibits better generalization than existing approximate second-order methods, we compare it with Sophia-H without lazy Hessian updating $k = 1$, ensuring that the algorithm is assessed under its optimal configuration.

## E.1. Image Classification

**CIFAR** We trained ResNet-20 and ResNet-32 on the CIFAR datasets for 160 epochs and Wide-ResNet28-10 for 200 epochs. Only standard inception-style data augmentations, such as random cropping and horizontal flipping, were applied, without any additional regularization techniques or extra augmentations. We used standard cross-entropy without label smoothing as a loss function. Also, we adopted a multi-step decay learning rate schedule. Specifically, for ResNet-20 and ResNet-32, the learning rate was decayed by a factor of 0.1 at epochs 80 and 120. For Wide-ResNet28-10, the learning rate was decayed by a factor of 0.2 at epochs 60, 120 and 160. The exponential moving average hyperparameters were set to $\beta_1 = 0.9$ and $\beta_2 = 0.999$. All experiments were conducted with a batch size of 256. The hyperparameter search space for each method is detailed in Table 9.

| Method | SASSHA | M-SASSHA | AdaHessian | Sophia-H | AdamW / SGD | SAM | shampoo |
|---|---|---|---|---|---|---|---|
| Learning Rate | $\left\{0.3, 0.15, 0.03, 0.015\right\}$ | | $\left\{0.3, 0.15, 0.1, 0.03, 0.015, 0.01, 0.003, 0.001, 0.0003, 0.0001\right\}$ | | | | $\left\{\begin{smallmatrix}1.5, 1.4, 1.3, 1.2, 1.1, 1, 0.9, 0.8, 0.7, 0.6, \\ 0.5, 0.4, 0.3, 0.2, 0.1, 0.01, 0.04, 0.004\end{smallmatrix}\right\}$ |
| Weight Decay | | | $\left\{\text{1e-3, 5e-4, 1e-4, 5e-5, 1e-5, 5e-6, 1e-6}\right\}$ | | | | |
| Perturbation radius $\rho$ | $\left\{0.1, 0.15, 0.2, 0.25\right\}$ | $\left\{0.1, 0.2, 0.3, 0.6, 0.8\right\}$ | - | - | - | $\left\{\begin{smallmatrix}0.01, 0.05, 0.1, 0.15, 0.2, 0.25, 0.3, \\ 0.35, 0.4, 0.45, 0.5, 0.55, 0.6\end{smallmatrix}\right\}$ | - |
| Clipping-threshold | - | - | - | $\left\{\begin{smallmatrix}0.1, 0.05, 0.01, 0.005, \\ 0.001, 0.0005, 0.0001\end{smallmatrix}\right\}$ | - | - | - |
| Damping | - | - | - | - | - | - | 1e-$\left\{2, 3, 4, 6, 8\right\}$ |
| Hessian Update Interval $k$ | 10 | 10 | 1 | 1 | - | - | 1 |
| learning rate schedule | | | Multi-step decay | | | | |

*Table 9.* Hyperparameter search space for CIFAR datasets

**ImageNet** We trained ResNet-50 and *plain Vision Transformer* (plain ViT) (Beyer et al., 2022) for 90 epochs. Remarkably, plain ViT converges in just 90 epochs on ImageNet, attaining performance comparable to the original ViT trained for 300 epochs (Beyer et al., 2022). This faster convergence allows us to efficiently assess whether SASSHA can enhance the generalization in ViT architectures. Consistent with our CIFAR training settings, we applied only standard inception-style data augmentations and used standard cross-entropy as a loss function. For ResNet-50, we adopted a multi-step decay learning rate schedule, reducing the learning rate by a factor of 0.1 at epochs 30 and 60. However, AdaHessian could not be trained with a multi-step decay schedule; therefore, as recommended by Yao et al. (2021), we employed a plateau decay schedule instead. For Vision Transformer training, following Chen et al. (2022), we used a cosine learning rate schedule with an 8-epoch warm-up phase. Additionally, the exponential moving average hyperparameters $\beta_1$ and $\beta_2$ were set to 0.9 and 0.999 respectively. We used a batch size of 256 for ResNet50 and 1024 for ViT. The hyperparameter search spaces for each methods used during training on the ImageNet dataset are detailed in Table 10.

## E.2. Language

**Language Pretraining** Following the training settings introduced in Gomes et al. (2024), we conducted experiments on a mini GPT-1 model using the Wikitext-2 dataset. This scaled-down version of GPT-1 maintains essential modeling capabilities while reducing computational demands. We trained the model with three methods: SASSHA, M-SASSHA, and Sophia-H. The hyperparameter tuning spaces for these methods are summarized in Table 11. For other methods not listed in the table, we directly reported the results from Gomes et al. (2024).

| methods | SASSHA | M-SASSHA | AdaHessian | Sophia-H | AdamW / SGD | SAM |
|---|---|---|---|---|---|---|
| Learning Rate | $\{0.6, 0.3, 0.15\}$ | $\{0.6, 0.3, 0.15\}$ | $\{0.6 0.3, 0.15\}$ | $\{0.4, 0.2, 0.1, 0.04, 0.02, 0.01, 0.001\}$ | | |
| Weight Decay | | | | $\{1e\text{-}3, 5e\text{-}4, 1e\text{-}4, 5e\text{-}5, 1e\text{-}5\}$ | | |
| Perturbation radius $\rho$ | $\{0.1, 0.15, 0.2, 0.25\}$ | $\{0.1, 0.2, 0.4, 0.8\}$ | - | - | - | $\{0.1, 0.15, 0.2, 0.25, 0.3\}$ |
| Clipping-threshold | - | - | - | $\{0.1, 0.05, 0.01, 0.005, 0.001, 0.0005, 0.0001\}$ | - | - |
| Hessian Update Interval $k$ | 10 | 10 | 1 | 1 | - | - |

*Table 10.* Hyperparameter search space for ImageNet

| methods | SASSHA / M-SASSHA | Sophia-H | SAM |
|---|---|---|---|
| Learning Rate | $\{0.15, 0.1, 0.03, 0.01, 0.003, 0.0015\}$ | $\{1e\text{-}2, 5e\text{-}3, 1e\text{-}3, 5e\text{-}4, 1e\text{-}4, 5e\text{-}5, 1e\text{-}5\}$ | $\{1e\text{-}2, 1e\text{-}3, 1e\text{-}4, 1e\text{-}5, 1e\text{-}6\}$ |
| Weight Decay | | $1e\text{-}\{1, 2, 3, 4, 5, 6, 7, 8\}$ | |
| Perturbation radius $\rho$ | $1e\text{-}\{1, 2, 3, 4, 5\}$ | - | $1e\text{-}\{1, 2, 3, 4, 5, 6, 7, 8\}$ |
| Clipping-threshold | - | $\{1e\text{-}1, 5e\text{-}2, 1e\text{-}2, 5e\text{-}3, 1e\text{-}3, 5e\text{-}4, 1e\text{-}4\}$ | - |
| Hessian Update Interval $k$ | 10 | 1 | - |
| Epochs | | 50 | 55 |

*Table 11.* Hyperparameter search space for language pretraining

**Language Finetuning**    We utilized a pretrained SqueezeBERT (Iandola et al., 2020) from the HuggingFace Hub (Wolf et al., 2020). We set the batch size to 16, the maximum sequence length to 512, and the dropout rate to 0. The number of training epochs varied depending on the specific GLUE task: 5 epochs for MNLI, QQP, QNLI, and SST-2; 10 epochs for STS-B, MRPC, and RTE. Additionally, We adopted a polynomial learning rate decay scheduler. The detailed hyperparameter search spaces are presented in Table 12.

| methods | SASSHA / M-SASSHA | Sophia-H | AdaHessian | AdamW | SAM |
|---|---|---|---|---|---|
| Learning Rate | | | $1e\text{-}\{1, 2, 3, 4, 5, 6, 7, 8\}$ | | |
| Weight Decay | | | $1e\text{-}\{1, 2, 3, 4, 5, 6, 7, 8\}$ | | |
| Perturbation radius $\rho$ | $1e\text{-}\{2, 3, 4, 5\}$ | - | - | - | $1e\text{-}\{1, 2, 3, 4, 5, 6, 7, 8\}$ |
| Clipping-threshold | - | $\left\{ \begin{matrix} 0.1, 0.05, 0.01, 0.005, \\ 0.001, 0.0005, 0.0001 \end{matrix} \right\}$ | - | - | |
| Hessian Update Interval $k$ | 1 | 1 | 1 | - | - |

*Table 12.* Hyperparameter search space for language finetuning

### E.3. Label Noise

We introduced label noise by randomly corrupting a fraction of the training data at rates of 20%, 40%, and 60%. Using this setup, we trained ResNet-32 for 160 epochs with a batch size of 256. We adopted a multi-step decay learning rate schedule, reducing the learning rate by a factor of 0.1 at epochs 80 and 120. The specific hyperparameters explored during these experiments are detailed in Table 13.

| Methods | SASSHA | M-SASSHA | Sophia-H | AdaHessian | SAM | SGD |
|---|---|---|---|---|---|---|
| Learning Rate | $\{0.3, 0.15, 0.1, 0.03, 0.015, 0.01, 0.003, 0.0015, 0.001\}$ | | | | | |
| Weight Decay | $\{1e\text{-}3, 5e\text{-}4, 5e\text{-}5, 1e\text{-}5, 5e\text{-}6, 1e\text{-}6\}$ | | | | | |
| Perturbation radius $\rho$ | $\{0.25, 0.2, 0.15, 0.1\}$ | $\{0.8, 0.6, 0.3, 0.2, 0.1\}$ | - | - | $\{0.3, 0.25, 0.2, 0.15, 0.1, 0.05, 0.02, 0.01, 0.002, 0.001\}$ | - |
| Clipping-threshold | - | - | $\{0.1, 0.05, 0.01, 0.005, 0.001, 0.0005, 0.0001\}$ | - | - | - |
| Hessian Update Interval $k$ | 10 | 10 | 1 | 1 | - | - |

*Table 13.* Hyperparameter search space for label noise experiments

# F. More Ablations

## F.1. Square Root Function

*Table 14.* Comparison of square root against damping and clipping.

| | CIFAR-10 | | CIFAR-100 |
|---|---|---|---|
| | ResNet-20 | ResNet-32 | ResNet-32 |
| Clipping | $92.78_{\pm0.18}$ | $93.80_{\pm0.16}$ | $69.47_{\pm0.20}$ |
| Damping | $92.74_{\pm0.06}$ | $93.68_{\pm0.29}$ | $71.27_{\pm0.43}$ |
| Square root (SASSHA) | $\mathbf{92.98}_{\pm0.05}$ | $\mathbf{94.09}_{\pm0.24}$ | $\mathbf{72.14}_{\pm0.16}$ |

We conduct an ablation study to support our use of the square rooted preconditioner in SASSHA, comparing it to other alternatives to stabilize the preconditioner such as damping or clipping. We search damping and clipping hyperparameters over $\{10^{-4}, 10^{-6}, 10^{-8}, 10^{-12}\}$ and $\{0.1, 0.05, 0.01, 0.005, 0.001, 0.0005, 0.0001\}$, respectively. We note that the square-root employed in SASSHA does not require such extensive hyperparamter search. The results are presented in Table 14.

Our experiments demonstrate that the square rooted preconditioner achieves higher validation accuracy than those with damping or clipping, even with a three times smaller hyperparameter search budget. We provide two possible explanations for this observation. First, taking square root preserves the relative scale among Hessian elements while smoothly amplifying near-zero entries in the denominator (*i.e.*, $h < \sqrt{h}$ if $0 < h < 1$). This property is particularly valuable during sharpness minimization, where the overall magnitude of the Hessian components tends to be small. In such cases, even small differences between Hessian elements may carry nontrivial curvature information. Applying Square root can help retain this relative structure while also reducing numerical instability caused by underestimated curvature. In contrast, both damping and clipping modify the Hessian by entirely shifting or abruptly replacing values based on a predefined and fixed threshold criterion. As a result, when the Hessian is generally small due to sharpness minimization, informative dimensions may fall below the threshold, removing potentially critical variations and hence deteriorating the quality of preconditioning. This behavior can also increase the sensitivity to the choice of the threshold hyperparameter. Second, square-rooted preconditioner can be interpreted as the result of a geometric interpolation between the identity matrix $I$ and $H^\alpha$. This interpolation has been demonstrated to enable selecting an optimal preconditioner that balances the bias and the variance of the population risk, thereby minimizing generalization error (Amari et al., 2021). In general, $\alpha = 1/2$ (i.e., square root) has consistently shown moderate performance across various scenarios (Amari et al., 2021; Duchi et al., 2011; Kingma & Ba, 2015).

## F.2. Absolute Value Function

We observe how the absolute function influences the training process to avoid convergence to a critical solution that could result in sub-optimal performance. We train ResNet-32 on CIFAR-100 using SASSHA without the absolute function (`No-Abs`) and compare the resulting training loss to that of the original SASSHA. We also plot the Hessian eigenspectrum of the found solution via the Lanczos algorithm ([Yao et al., 2020](#)) to determine whether the found solution corresponds to a minimum or a saddle point. The results are illustrated in Figure 6. We can see that without the absolute function, the training loss converges to a sub-optimal solution, where the prevalent negative values in the diagonal Hessian distribution indicate it as a saddle point. This shows the necessity of the absolute function for preventing convergence to these critical regions.

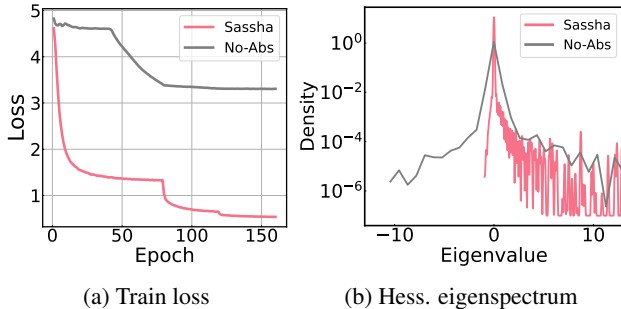

(a) Train loss      (b) Hess. eigenspectrum

*Figure 6.* Effect of the absolute function on the training loss and the Hessian eigenspectrum of the found solution of SASSHA on ResNet-32/CIFAR-10. Without the absolute function, SASSHA converges to sub-optimal saddle point.

## G. Validation Loss Curve for Vision Task

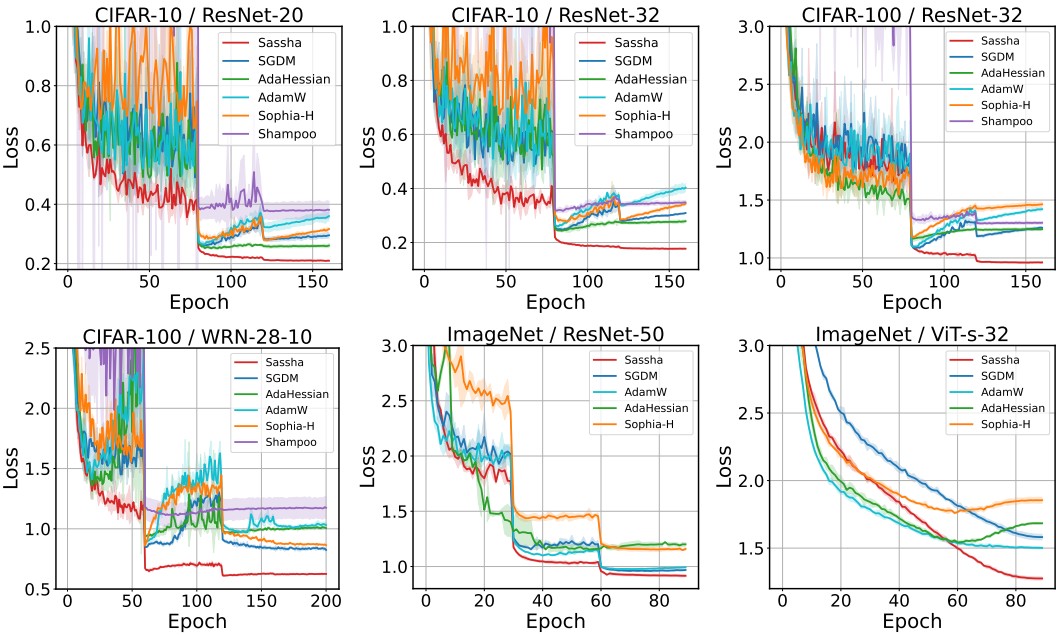

*Figure 7.* Validation loss curve of SASSHA, SGD, AdaHessian, AdamW, and Sophia-H on various image classification models and tasks. SASSHA outperforms all first-order and second-order baseline methods.

The experimental results in Figure 7 demonstrate better generalization capability of SASSHA over the related methods. Across all datasets and model architectures, our method consistently achieves the lowest validation loss, indicative of its enhanced ability to generalize from training to validation data effectively. This robust performance of SASSHA underscores its potential as a leading optimization method for various deep learning applications, particularly in image classification.

## H. Comparison with First-order Baselines with Given More Training Budget than SASSHA

We train SGD and AdamW for twice as many epochs as SASSHA and compare their final validation accuracies. The results are presented in Table 15. Despite this extended training budget, these first-order methods fall short of the performance attained with SASSHA, demonstrating their limited effectiveness compared to SASSHA. We attribute this outcome to SASSHA reaching a flatter and better generalizing solution along with stable preconditioning, which together enables consistent outperformance over first-order baselines.

*Table 15.* Performance comparison of SASSHA against SGD and AdamW with twice the epoch allocation. SASSHA achieves better results with significantly fewer epochs.

|  | RN20 - CIFAR-10
Acc (epoch) | RN32 - CIFAR-10
Acc (epoch) | RN32 - CIFAR-100
Acc (epoch) | WRN28 - CIFAR-100
Acc (epoch) | RN50 - ImageNet
Acc (epoch) | ViT_s - ImageNet
Acc (epoch) |
|---|---|---|---|---|---|---|
| SGD | 92.62 (320e) | 93.43 (320e) | 69.93 (320e) | 80.50 (400e) | 75.90 (180e) | 63.64 (180e) |
| AdamW | 92.55 (320e) | 92.97 (320e) | 69.50 (320e) | 79.46 (400e) | 75.57 (180e) | 66.97 (180e) |
| SASSHA | **92.98** (160e) | **94.09** (160e) | **72.14** (160e) | **83.54** (200e) | **76.43** (90e) | **69.20** (90e) |

## I. Effectiveness of Stable Hessian Approximations in SASSHA

*Table 16.* Results of Sophia-H with sharpness minimization.

|  | CIFAR-10 | | CIFAR-100 | |
|---|---|---|---|---|
|  | ResNet-20 | ResNet-32 | ResNet-32 | WRN-28-10 |
| SAM | $92.85_{\pm 0.07}$ | $93.89_{\pm 0.13}$ | $71.99_{\pm 0.20}$ | $83.14_{\pm 0.13}$ |
| Sophia-H (with SAM) | $92.53_{\pm 0.39}$ | $93.59_{\pm 0.31}$ | $71.31_{\pm 0.43}$ | $80.15_{\pm 0.35}$ |
| SASSHA | $\mathbf{92.98}_{\pm 0.05}$ | $\mathbf{94.09}_{\pm 0.24}$ | $\mathbf{72.14}_{\pm 0.16}$ | $\mathbf{83.54}_{\pm 0.08}$ |

We demonstrate limited benefit from naively combining SAM with existing approximate second-order methods without the carefully designed stabilization strategies of SASSHA. Precisely, we compare the validation accuracy of SASSHA with a simple combination of SAM and Sophia, denoted as Sophia-H (with SAM). We provide results in Table 16.

We observe that Sophia-H (with SAM) performs worse than SAM, whereas SASSHA outperforms both methods, validating the effectiveness of the design choices made in SASSHA. We attribute this to the reduced compatibility of Sophia-H with SAM compared to SASSHA. First, Sophia clipping destroys the relative scale between individual elements of the Hessian. This may be particularly problematic when using SAM, where the overall Hessian values tend to be small. In such cases, even very small differences between Hessian elements may carry nontrivial curvature information. However, Sophia clipping abruptly replaces small or negative Hessian values based on a predefined and fixed threshold criterion. As a result, when the Hessian is generally small due to SAM, informative dimensions may fall below the threshold, removing potentially critical variations and thereby deteriorating the quality of preconditioning. This situation also raises the sensitivity to hyperparameters like the clipping threshold and makes the optimization process more dependent on careful tuning. Conversely, the stable Hessian approximation in SASSHA, incorporating the absolute function and square rooting, preserves the relative scale among the Hessian entries by smoothly adjusting their magnitudes.

In addition, the use of sophia clipping results in Sophia partially performing signSGD over a subset of parameters (Liu et al., 2024), which may lead to suboptimal convergence in typical situations (Karimireddy et al., 2019).

## J. Additional Language Modeling Experiments

In this section, we provide additional language modeling experiments. We train GPT2-small for 50k iterations on OpenWebText (Gao et al., 2020), using various optimization methods for comparison. For AdamW and Sophia-G, we adopt the best hyperparameter configurations reported by (Liu et al., 2024), and set $\rho = 0.1$ for all SAM-related methods. Due to limited computational resources, all experiments were run with a single random seed. The results are presented in Table 17. We observe that SASSHA achieves performance comparable to related methods.

Table 17. GPT2 pretraining results. SASSHA achieves similar performance to state-of-the-art methods.

| Method | Loss | Perplexity |
|---|---|---|
| AdamW | 2.9622 | 19.353 |
| Sophia-G | 2.9307 | 18.751 |
| SAM $_{AdamW}$ | 2.9558 | 19.196 |
| Sophia-G (with SAM) | 2.9319 | 18.773 |
| SASSHA | 2.9445 | 19.015 |

## K. Comparison with Advanced SAM Variants

Thus far, our primary focus has centered on validating the effectiveness of SASSHA in the context of approximate second-order optimization. While this remains the principal objective of our study, here we additionally compare SASSHA with advanced SAM variants (*i.e.* ASAM (Kwon et al., 2021), GSAM (Zhuang et al., 2022)) to prove that SASSHA is a sensible approach. We also evaluate G-SASSHA (SASSHA with surrogate gap guided sharpness from (Zhuang et al., 2022)) for fair comparison. The results are represented in Table 18.

Table 18. SASSHA v.s. advanced SAM variants in Image classification.

| | CIFAR-10 | | CIFAR-100 | | ImageNet | |
|---|---|---|---|---|---|---|
| | ResNet-20 | ResNet-32 | ResNet-32 | WRN-28-10 | ResNet-50 | ViT-s-32 |
| ASAM | $92.96_{\pm 0.25}$ | $93.85_{\pm 0.15}$ | $72.02_{\pm 0.28}$ | $83.39_{\pm 0.06}$ | $76.54_{\pm 0.15}$ | $68.26_{\pm 0.36}$ |
| GSAM | $92.72_{\pm 0.39}$ | $93.76_{\pm 0.31}$ | $72.10_{\pm 0.43}$ | $83.21_{\pm 0.39}$ | $76.45_{\pm 0.22}$ | $69.60_{\pm 0.16}$ |
| SASSHA | $\mathbf{92.98}_{\pm 0.05}$ | $94.09_{\pm 0.24}$ | $72.14_{\pm 0.16}$ | $83.54_{\pm 0.08}$ | $76.43_{\pm 0.18}$ | $69.20_{\pm 0.30}$ |
| G-SASSHA | $92.94_{\pm 0.18}$ | $\mathbf{94.15}_{\pm 0.12}$ | $\mathbf{72.18}_{\pm 0.52}$ | $\mathbf{83.56}_{\pm 0.27}$ | $\mathbf{76.66}_{\pm 0.23}$ | $\mathbf{69.67}_{\pm 0.14}$ |

We find that SASSHA is competitive with these advanced SAM variants. However, we note clearly that those SAM variants require considerably more hyperparameter tuning to achieve generalization performance comparable to SASSHA. For example, GSAM introduces an additional hyperparameter $\alpha$, demanding as much tuning effort as tuning $\rho$. Similarly, ASAM, as noted by its authors, typically necessitates exploring a broader $\rho$ range, as its appropriate value is approximately 10 times larger than that of SAM. In our setup, tuning GSAM and ASAM involved $4.5\times \sim 15.75\times$ and $3\times \sim 8\times$ larger search grids compared to SASSHA, respectively. We provide detailed setup and hyperparameter search space below.

**Setup and Search space.** For ResNet, we use SGD as the base methods for ASAM and GSAM, while for ViT, AdamW with gradient clipping set to 1.0 serves as the base methods. For all models, typical cross entropy loss is used (not label-smoothing cross entropy), and the best learning rate and weight decay of the base methods are selected in experiments with ASAM and GSAM. All algorithms are evaluated with constant $\rho$ (without scheduling). For learning rate schedule, we apply multi-step decay with a decay rate of 0.1 for ResNet on CIFAR, and use cosine learning rate decay with 8 warm-up epochs for ViTs.

| | ResNet/CIFAR | ResNet/ImageNet | ViT/ImageNet |
|---|---|---|---|
| $\rho$ | $\{0.01, 0.05, 0.1, 0.15, 0.2, 0.25, 0.3, 0.35, 0.4\}$ | $\{0.01, 0.05, 0.1, 0.15, 0.2, 0.25, 0.3\}$ | $\{0.1, 0.2, 0.3, 0.4, 0.5, 0.6\}$ |
| $\alpha$ | $\{0.01, 0.05, 0.1, 0.15, 0.2, 0.25, 0.3\}$ | $\{0.01, 0.05, 0.1, 0.15, 0.2\}$ | $\{0.1, 0.2, 0.3\}$ |

Table 19. Hyperparameter search space for GSAM and G-SASSHA

| | ResNet/CIFAR | ImageNet |
|---|---|---|
| $\rho$ | $\left\{\begin{array}{l}0.01,0.05,0.1,0.15,0.2,0.25,0.3,0.4,0.5,0.6,0.7,0.8,0.9,1,\\1.1,1.2,1.3,1.4,1.5,1.6,1.7,1.8,1.9,2,2.5,3,3.5,4,4.5,5,5.5,6\end{array}\right\}$ | $\{0.1, 0.15, 0.2, 0.3, 0.4, 0.5, 0.6, 0.7, 0.8, 0.9, 1, 1.5, 2\}$ |

*Table 20.* Hyperparameter search space for ASAM

## L. Additional Label Noise Experiments

*Table 21.* Robustness to label noise. Here we measure the validation accuracy under various levels of label noise using ResNet-32 trained on CIFAR-100 and CIFAR-10. SASSHA shows much robust performance under label noise.

| | CIFAR-10 | | | | CIFAR-100 | | | |
|---|---|---|---|---|---|---|---|---|
| Noise level | 0% | 20% | 40% | 60% | 0% | 20% | 40% | 60% |
| SGD | $92.69_{\pm0.06}$ | $89.91_{\pm0.87}$ | $87.26_{\pm0.40}$ | $82.72_{\pm1.59}$ | $69.32_{\pm0.19}$ | $62.18_{\pm0.06}$ | $55.78_{\pm0.55}$ | $45.53_{\pm0.78}$ |
| SAM $_{\text{SGD}}$ | $93.89_{\pm0.13}$ | $92.27_{\pm0.14}$ | $90.11_{\pm0.25}$ | $85.79_{\pm0.30}$ | $71.99_{\pm0.20}$ | $65.53_{\pm0.11}$ | $61.20_{\pm0.17}$ | $51.93_{\pm0.47}$ |
| AdaHessian | $92.48_{\pm0.15}$ | $90.11_{\pm0.01}$ | $86.88_{\pm0.04}$ | $83.25_{\pm0.01}$ | $68.06_{\pm0.22}$ | $63.06_{\pm0.25}$ | $58.37_{\pm0.13}$ | $46.02_{\pm1.96}$ |
| Sophia-H | $91.99_{\pm0.08}$ | $89.93_{\pm0.01}$ | $87.30_{\pm0.51}$ | $82.78_{\pm1.43}$ | $67.76_{\pm0.37}$ | $62.34_{\pm0.47}$ | $56.54_{\pm0.28}$ | $45.37_{\pm0.27}$ |
| Shampoo | $90.23_{\pm0.83}$ | $88.14_{\pm0.29}$ | $85.15_{\pm0.61}$ | $81.16_{\pm0.30}$ | $64.08_{\pm0.46}$ | $58.85_{\pm0.66}$ | $53.82_{\pm0.71}$ | $42.91_{\pm0.99}$ |
| SASSHA | $\mathbf{94.09}_{\pm0.24}$ | $\mathbf{92.49}_{\pm0.11}$ | $\mathbf{90.29}_{\pm0.11}$ | $\mathbf{86.50}_{\pm0.08}$ | $\mathbf{72.14}_{\pm0.16}$ | $\mathbf{66.78}_{\pm0.47}$ | $\mathbf{61.97}_{\pm0.27}$ | $\mathbf{53.98}_{\pm0.57}$ |

## M. M-SASSHA: Efficient Perturbation

Having explored techniques to reduce the computational cost of second-order methods, here we consider employing techniques to alleviate the additional gradient computation in sharpness-minimization. Prior works have suggested different ways to reduce this computational overhead including infrequent computations (Liu et al., 2022), use of sparse perturbations (Mi et al., 2022), or computing with selective weight and data (Du et al., 2022a). In particular, we employ the approaches of Becker et al. (2024), which uses the normalized negative momentum as the perturbation:

$$\epsilon_t^\star = \rho \frac{m_{t-1}}{\|m_{t-1}\|_2}, \tag{17}$$

which entirely eliminates the need for additional gradient computation with similar generalization improvement as the original SAM. We call this low-computation alternative as M-SASSHA and evaluate this across vision, language, and label noise tasks, as we did in the main sections. The results are presented in Tables 22 to 24, respectively.

Despite having a computational cost comparable to first-order methods like SGD and Adam, and significantly lower than approximate second-order methods, M-SASSHA demonstrates superior performance over both first-order and second-order approaches. In image classification, M-SASSHA proves more effective than the best-performing approximate second-order methods by 2% on CIFAR-100 with ResNet-32 and by 2.5% with ResNet-50, while also exceeding AdamW by approximately 1.6% on ViT. For language pretraining, it attains a test perplexity that is 22 points lower than the second-best performinig Sophia-H and outperforms AdamW in nearly all language tasks. Lastly, M-SASSHA surpasses other methods across all noise levels, proving highly resilient in the presence of extreme label noise. These results reaffirm the effectiveness and consistency of our well-engineered design choices, which enable the stable integration of efficient sharpness minimization into second-order optimization while retaining its benefits.

*Table 22.* M-SASSHA v.s. baselines in image classification. M-SASSHA shows superior performance.

| Category | Method | CIFAR-10 | | CIFAR-100 | | ImageNet | |
|---|---|---|---|---|---|---|---|
| | | ResNet-20 | ResNet-32 | ResNet-32 | WRN-28-10 | ResNet-50 | ViT-s-32 |
| First-order | SGD | $92.03_{\pm0.32}$ | $92.69_{\pm0.06}$ | $69.32_{\pm0.19}$ | $80.06_{\pm0.15}$ | $75.58_{\pm0.05}$ | $62.90_{\pm0.36}$ |
| | AdamW | $92.04_{\pm0.11}$ | $92.42_{\pm0.13}$ | $68.78_{\pm0.22}$ | $79.09_{\pm0.35}$ | $75.38_{\pm0.08}$ | $66.46_{\pm0.15}$ |
| Second-order | AdaHessian | $92.00_{\pm0.17}$ | $92.48_{\pm0.15}$ | $68.06_{\pm0.22}$ | $76.92_{\pm0.26}$ | $73.64_{\pm0.16}$ | $66.42_{\pm0.23}$ |
| | Sophia-H | $91.81_{\pm0.27}$ | $91.99_{\pm0.08}$ | $67.76_{\pm0.37}$ | $79.35_{\pm0.24}$ | $72.06_{\pm0.49}$ | $62.44_{\pm0.36}$ |
| | Shampoo | $88.55_{\pm0.83}$ | $90.23_{\pm0.24}$ | $64.08_{\pm0.46}$ | $74.06_{\pm1.28}$ | * | * |
| | M-SASSHA | $\mathbf{92.36}_{\pm0.23}$ | $\mathbf{93.18}_{\pm0.30}$ | $\mathbf{70.93}_{\pm0.21}$ | $\mathbf{81.53}_{\pm0.27}$ | $\mathbf{76.00}_{\pm0.04}$ | $\mathbf{68.04}_{\pm0.14}$ |

*Table 23.* M-SASSHA v.s. baselines in language tasks. For pretraining, M-SASSHA achieves the lowest perplexity among all methods. For finetuning, M-SASSHA performs better than AdamW and compares competitively with Sophia-H.

| | Pretrain / GPT1-mini | Finetune / SqeezeBERT | | | | | | |
|---|---|---|---|---|---|---|---|---|
| | Wikitext-2 Perplexity | SST-2 Acc | MRPC Acc / F1 | STS-B S/P corr. | QQP F1 / Acc | MNLI mat/m.mat | QNLI Acc | RTE Acc |
| AdamW | 175.06 | $90.29_{\pm0.52}$ | $84.56_{\pm0.25}$ / $88.99_{\pm0.11}$ | $88.34_{\pm0.15}$ / $88.48_{\pm0.20}$ | $89.92_{\pm0.05}$ / $86.58_{\pm0.11}$ | $81.22_{\pm0.07}$ / $82.26_{\pm0.05}$ | $89.93_{\pm0.14}$ | $68.95_{\pm0.72}$ |
| AdaHessian | 407.69 | $89.64_{\pm0.13}$ | $79.74_{\pm4.00}$ / $85.26_{\pm3.50}$ | $86.08_{\pm4.04}$ / $86.46_{\pm4.06}$ | $90.37_{\pm0.05}$ / $87.07_{\pm0.05}$ | $81.33_{\pm0.17}$ / $82.08_{\pm0.02}$ | $89.94_{\pm0.12}$ | $\mathbf{71.00}_{\pm1.04}$ |
| Sophia-H | 157.60 | $\mathbf{90.44}_{\pm0.46}$ | $85.78_{\pm1.07}$ / $89.90_{\pm0.82}$ | $88.17_{\pm1.07}$ / $\mathbf{88.53}_{\pm1.13}$ | $90.70_{\pm0.04}$ / $87.60_{\pm0.06}$ | $\mathbf{81.77}_{\pm0.18}$ / $\mathbf{82.36}_{\pm0.22}$ | $\mathbf{90.12}_{\pm0.14}$ | $70.76_{\pm1.44}$ |
| M-SASSHA | $\mathbf{125.01}$ | $90.332_{\pm0.88}$ | $\mathbf{87.092}_{\pm1.98}$ / $\mathbf{90.599}_{\pm1.51}$ | $\mathbf{88.37}_{\pm0.04}$ / $88.46_{\pm0.07}$ | $\mathbf{90.78}_{\pm0.05}$ / $\mathbf{87.61}_{\pm0.07}$ | $81.42_{\pm0.19}$ / $81.94_{\pm0.09}$ | $89.84_{\pm0.22}$ | $70.40_{\pm0.96}$ |

*Table 24.* Robustness to label noise. Here we measure the validation accuracy under various levels of label noise using ResNet-32 trained on CIFAR-100 and CIFAR-10. M-SASSHA shows much robust performance under label noise.

| Noise level | CIFAR-10 | | | | CIFAR-100 | | | |
|---|---|---|---|---|---|---|---|---|
| | 0% | 20% | 40% | 60% | 0% | 20% | 40% | 60% |
| SGD | $92.69_{\pm0.06}$ | $89.91_{\pm0.87}$ | $87.26_{\pm0.40}$ | $82.72_{\pm1.59}$ | $69.32_{\pm0.19}$ | $62.18_{\pm0.06}$ | $55.78_{\pm0.55}$ | $45.53_{\pm0.78}$ |
| AdaHessian | $92.48_{\pm0.15}$ | $90.11_{\pm0.01}$ | $86.88_{\pm0.04}$ | $83.25_{\pm0.01}$ | $68.06_{\pm0.22}$ | $63.06_{\pm0.25}$ | $58.37_{\pm0.13}$ | $46.02_{\pm1.96}$ |
| Sophia-H | $91.99_{\pm0.08}$ | $89.93_{\pm0.01}$ | $87.30_{\pm0.51}$ | $82.78_{\pm1.43}$ | $67.76_{\pm0.37}$ | $62.34_{\pm0.47}$ | $56.54_{\pm0.28}$ | $45.37_{\pm0.27}$ |
| Shampoo | $90.23_{\pm0.83}$ | $88.14_{\pm0.29}$ | $85.15_{\pm0.61}$ | $81.16_{\pm0.30}$ | $64.08_{\pm0.46}$ | $58.85_{\pm0.66}$ | $53.82_{\pm0.71}$ | $42.91_{\pm0.99}$ |
| M-SASSHA | $\mathbf{93.18}_{\pm0.23}$ | $\mathbf{91.27}_{\pm0.31}$ | $\mathbf{88.85}_{\pm0.31}$ | $\mathbf{85.17}_{\pm0.24}$ | $\mathbf{70.93}_{\pm0.21}$ | $\mathbf{66.10}_{\pm0.26}$ | $\mathbf{61.13}_{\pm0.28}$ | $\mathbf{52.45}_{\pm0.34}$ |

