# OpenReview forum: "Sassha: Sharpness-aware Adaptive Second-order Optimization with Stable Hessian Approximation"
_ICML.cc/2025/Conference — ICML 2025 poster_

### Official Review · Reviewer_VPX6 · 2025-03-06

**Overall Recommendation:** 2

**Summary:**

Combine diag hessian estimate optimizer with SAM/M-SAM

**Claims And Evidence:**

The paper is written clearly. It combines an adaptive(adamized) diagonal 2nd order optimizer with M-SAM. This results in an empirical boost over other optimizers that.

**Essential References Not Discussed:**

This paper is basically just a different version of Flat Sophia proposed last year https://github.com/evanatyourservice/flat-sophia

**Experimental Designs Or Analyses:**

The experiment baselines did not always make sense. For example, ViTs should be optimized with Adam or Shampoo not with SGD. So that experiment is a bit moot. Also a very baisic resnet can get 94%

**Methods And Evaluation Criteria:**

For imagenet ViT SAM is not SoTA. One should be using Shampoo or Adam. Also SGD is not good for optimizing transformers so the baselines look a bit skewed.

**Other Comments Or Suggestions:**

NA

**Other Strengths And Weaknesses:**

I am very happy the authors are using Momentum SAM which is basically just SAM but without the extra back prop. https://arxiv.org/abs/2401.12033

I also very much like that they provided the sharpness comparison of the momentum/full sam version of the optimizer.

I dont think combining M-SAM and diagonal 2nd order is novel enough.

**Questions For Authors:**

NA

**Relation To Broader Scientific Literature:**

This paper is basically just a different version of Flat Sophia proposed last year https://github.com/evanatyourservice/flat-sophia

In RL many people have combined SAM and 2nd order opts to boost neuro plasticity. This paper is not novel in this.

**Theoretical Claims:**

NA

---

> ### Author Rebuttal · Authors · 2025-04-01
>
> We really appreciate the reviewer’s feedback. While we address the reviewer’s specific comments below, we would be keen to engage in any further discussion.
>
> ---
>
> > “ViTs should be optimized with Adam or Shampoo not with SGD”
>
> There seems to be some confusion. We have already provided results for ViT trained with AdamW in Table 2.
>
> ---
>
> > “a very baisic resnet can get 94%”
>
> We would appreciate it if you could provide a specific reference for the claimed 94% performance of a basic ResNet. The paper we are referencing [1] reports results similar to ours.
>
> ---
>
> > “This paper is basically just a different version of Flat Sophia” [2]
>
> Thank you for providing a pointer. With all due respect, however, we disagree with this assessment and find it to be quite stretched, if not unfair.
>
> First, Sassha operates very differently from Flat-Sophia, notably in its sharpness minimization approach [3,4] and second-order techniques (see Appendix B), and such differences are known to yield distinct optimization behavior [5].
>
> More importantly, however, Sassha is supported by a comprehensive empirical and theoretical study that offers insights and presented as a form of research paper, rendering potentially greater value in a significantly more reliable fashion compared to the suggested pointer, which seems to be a code repository in an idea level without providing any numerical result  [2].
>
> We sincerely hope that the reviewer sees the value we created in this work to architect a generally well-performing method for standard deep learning settings.
>
>
> ---
>
> > “In RL many people have combined SAM and 2nd order opts to boost neuro plasticity. This paper is not novel in this.”
>
> Thank you for your comment. We would appreciate it if you could point us to specific references. This would allow us to more accurately compare under fair settings and discuss their relations to our work.
>
> In the meantime, we hope that the reviewer sees the contributions we made in this work given that the method is rigorously evaluated across diverse standard settings to set a new state of the art, and that leveraging general ideas (i.e., sharpness minimization and second-order optimization) should not necessarily be used as a ground for criticism.
>
>
>
> &nbsp;
>
> **Reference**
>
> [1] Yao et al., ADAHESSIAN: An Adaptive Second Order Optimization for Machine Learning, AAAI, 2021.\
> [2] https://github.com/evanatyourservice/flat-sophia. \
> [3] Wang et al., Improving Generalization and Convergence by Enhancing Implicit Regularization, NeurIPS 2024.\
> [4] Foret et al., Sharpness-Aware Minimization for Efficiently Improving Generalization, ICLR, 2021.\
> [5] Dauphin et al., Neglected Hessian component explains mysteries in sharpness regularization., NeurIPS 2024.

---

> > ### Comment · Reviewer_VPX6 · 2025-04-04
> >
> > > There seems to be some confusion. We have already provided results for ViT trained with AdamW in Table 2.
> >
> > In table 2 SAM_Adamw is the next best reported optimizer for ViT CF10. I am saying SAM based optimizers are not SoTA for ViT's so this leads me to believe that the baseline is not tuned.
> >
> > > We would appreciate it if you could provide a specific reference for the claimed 94% performance of a basic ResNet. The paper we are referencing [1] reports results similar to ours.
> >
> > This resnet style model hits 94% in 2 around seconds and 96% in a few more seconds.
> >
> > https://github.com/KellerJordan/cifar10-airbench/blob/master/legacy/airbench94.py
> >
> > If one really wants one can go back in the commits and Keller had a Resnet 18 using super convergence that hit 94% in ~10-15 epochs. Here is something I coded up very quickly that hits 94% in 26 epochs with vanilla SGD + M.  https://codeshare.io/XLeWEE
> >
> >
> > > “This paper is basically just a different version of Flat Sophia” [2]
> >
> > I stand by this claim. I certainly respect the hard work the authors have done and of course understand and appreciate the rigor of academic work. I was simply saying that this paper proposes a different way of doing ideas behind flat Sophia. In essence it's the blend of sharpness/flatness aware minimization as well as 2nd order methods. There was no intent to be reductive.
> >
> >
> > > “In RL many people have combined SAM and 2nd order opts to boost neuro plasticity. This paper is not novel in this.”
> > I will have to look for the work/github code, but I know firsthand of people that have done CASPR (Shampoo's big brother) + SAM before in RL.
> >
> > I think in general SAM and 2nd order are two orthogonal frameworks putting them together is natural but is not novel for a publication.

---

> > > ### Author Response · Authors · 2025-04-06
> > >
> > > >  “SAM based optimizers are not SoTA for ViT's so this leads me to believe that the baseline is not tuned.”
> > >
> > > - First of all, `SAM_AdamW` should be considered an enhanced version of `AdamW` since it is simply `SAM` with `AdamW` as the base minimizer; i.e., `SAM_AdamW` subsumes `AdamW`. Therefore, **it is very natural to see that `SAM_AdamW` performs better than AdamW, as in not only in our experiments, but also in many other prior work especially on ViT [1-3]**. In fact, `SAM_AdamW` outperforming `AdamW` proves that `SAM_AdamW` is tuned well.
> > > - Also, we assure the reviewer that `AdamW` is tuned properly too, and this results align well with other reports in a similar setup [3-4].
> > > - Please note that we have performed rigorous hyperparameter tuning for *all methods* reported in this work, which can be reproduced through details provided in Appendix D.
> > >
> > > ---
> > > > “This resnet style model hits 94% in 2 around seconds and 96% in a few more seconds. …”
> > >
> > > Thank you for providing specific references [5-6]. However, **their experimental settings deviate substantially from ours (and those commonly used in the literature [7-9])** as below:
> > >
> > > |-|Git Repo [5]|Code [6]|Ours|
> > > |-|-|-|-|
> > > |Architecture|Customized CNN (**1.97M**)|ResNet18 (**11.17M**)|ResNet32 (**0.47M**)|
> > > |Test-time multiple input augmentation|O (crop)|O (flip)|X|
> > > |Activation function|GELU|RELU|RELU|
> > > |Label smoothing|O|X|X|
> > > |LR scheduler|Triangular|Triangular|Basic step decay|
> > > |2-pixel random translation with reflection padding|O|X|X|
> > > |Initialization|Frozen patch-whitening + Identity initialization|Standard|Standard|
> > > |Optimization tricks|O (scale bias + lookahead)|X|X|
> > > |Advanced augmentation|O (ALTflip)|O (Cutout, translation)|X|
> > >
> > > As such, **making a direct comparison would not be fair, nor would adopting such settings be relevant or necessary for the purpose of this work**. We also emphasize that **all methods reported in our paper were evaluated under the same setting** so as to verify the effects of the proposed ideas in a fair, transparent, and standard setting.
> > >
> > > ---
> > > > “SAM and 2nd order are two orthogonal frameworks putting them together is natural but is not novel for a publication.”
> > >
> > > The reviewer may have misunderstood a core aspect of our work. **The two mechanisms are *NOT* orthogonal at all**. Specifically, sharpness minimization reduces curvature, which directly affects the preconditioning of second-order methods. Sassha is a well-architected method that mitigates this *strongly-associated* issue of Hessian underestimation that occurs in this process, while at the same time allowing lazy Hessians to be accommodated, making it not only stable and efficient, but also generalizable (and much better than other practical second-order methods).
> > >
> > > In contrast, **simply “putting them together” performs worse compared to Sassha**. For instance, flat sophia, which is simply IRE[10] + Sophia, underperforms Sassha on training ResNet32 on CIFAR-10:
> > >
> > > ||val accuracy|
> > > |-|-|
> > > |flat-sophia|93.24|
> > > |sassha|**94.09**|
> > >
> > > Likewise, simple SAM+Sophia underperforms Sassha too (see Appendix G).
> > >
> > > Nevertheless, we acknowledge that this perspective may not have been clearly communicated in the paper, and we will revise the final version.
> > >
> > > ---
> > > >  “I will have to look for the work/github code … 2nd opt+SAM in RL ”
> > >
> > > We kindly request the reviewer to provide us with a specific reference point on this. Although we are still not quite convinced as to why Sassha has to be compared with methods developed for RL, we would be keen to address them further.
> > >
> > > ---
> > > > “I certainly respect the hard work the authors have done and of course understand and appreciate the rigor of academic work.”
> > >
> > > We sincerely appreciate the reviewer’s kind words and recognition of our efforts. In light of your positive remarks regarding the effort and rigor of our work, we kindly ask whether you might be open to reconsidering the current score. We believe the contributions presented—both in terms of methodology and analysis—offer meaningful value to the community, and we hope they align with the standards you associate with a higher evaluation. We are, of course, grateful for your careful consideration regardless of the outcome and will reflect your suggestions on the final version.
> > >
> > > ---
> > > **References** \
> > > [1] Chen et al., When Vision Transformers Outperform ResNets without Pre-training or Strong Data Augmentations \
> > > [2] Liu et al., Towards efficient and scalable sharpness-aware minimization \
> > > [3] Beyer et al., Better plain ViT baselines for ImageNet-1K \
> > > [4] How to train your ViT? Data, Augmentation, and Regularization in Vision Transformers \
> > > [5] https://github.com/KellerJordan/cifar10-airbench/blob/master/legacy/airbench94.py \
> > > [6] https://codeshare.io/XLeWEE \
> > > [7] He et al., Deep residual learning for image recognition \
> > > [8] Zagoruyko et al., Wide residual networks \
> > > [9] Yao et al., ADAHESSIAN \
> > > [10] Wang et al., Improving Generalization and Convergence by Enhancing Implicit Regularization

---

### Official Review · Reviewer_Wdix · 2025-03-07

**Overall Recommendation:** 3

**Summary:**

This paper compares the sharpness and generalisation of solutions found by second-order vs first-order optimisers, observing worse generalisation and larger sharpness of second-order methods.
To rescue test performance of second-order optimisers, it proposes an optimization method combining (diagonal) second-order optimization with sharpness-aware minimisation, obtaining better generalisation than first-order (including adaptive) methods.
A few crucial design choices are made to stabilise training, such as taking the absolute value and square root of the diagonal of the Hessian.
It is shown that the Hessian changes during training more slowly with respect to other optimisers, allowing infrequent Hessian updates, that are computationally convenient.
The method is tested on image classification and language modelling.

**Claims And Evidence:**

The debate on whether second-order methods generalise is ongoing, and this paper provides a significant contribution.

The combination of second-order optimization with sharpness-aware minimisation seems novel.

Empirical results are compelling.

Some of the design choices are not well justified, it is really not clear why square rooting should be better than damping and/or clipping.

The section “Comparison with SAM” is not quite convincing in general. The authors claim that SASSHA is more robust to block heterogeneity inherent in Transformer architectures. Then, they show that SAM underperforms SASSHA, even with more training iterations. I don’t see how this result relates with the claim.

**Essential References Not Discussed:**

OK

**Experimental Designs Or Analyses:**

OK

**Methods And Evaluation Criteria:**

SAM gets 79.1 accuracy with ResNet50 on ImageNet, while the authors report 76.3. Is it because of differences in data augmentation?

Sam also gets 83.7 accuracy with WRN-28-10 on CIFAR-100, using basic augmentation, so that should be the same used by the authors. However, the authors report 82.9. While the difference is small, it would reverse the claim that SASSHA, which gets 83.5, has a better performance.

**Other Comments Or Suggestions:**

NA

**Other Strengths And Weaknesses:**

NA

**Questions For Authors:**

NA

**Relation To Broader Scientific Literature:**

OK

**Theoretical Claims:**

NA

---

> ### Author Rebuttal · Authors · 2025-04-01
>
> We’re sincerely grateful for the reviewer’s thoughtful engagement and recognition of our contribution. It was encouraging and helped us further refine the work. We provide our responses below and welcome further discussion.
>
> ---
>
> **Justification for design choices**
>
> Thank you for your comment. We believe square-rooting can be potentially more effective than clipping or damping for two main reasons.
> First, square-rooting preserves the relative scale between each element of the Hessian. This property is particularly valuable when the sharpness minimization is underway, where the overall Hessian values tend to be small. In such cases, even small differences between Hessian elements may carry nontrivial curvature information. Square-rooting can help retain this relative structure while also mitigating numerical instability caused by underestimated curvature. In contrast, both clipping and damping operate by abruptly replacing Hessian values based on a predefined and fixed threshold criterion. As a result, when the Hessian is generally small due to sharpness minimization, informative dimensions may fall below the threshold,  removing potentially critical variations and hence deteriorating the quality of preconditioning. This behavior can also make the method more sensitive to hyperparameter choices.
>
> Second, square-rooting can further be interpreted as a geometric interpolation between a Hessian-based preconditioner and the identity matrix, which, as theoretically analyzed in [1], provides a natural mechanism for balancing between bias and variance.
>
> Additionally, clipping mechanism in Sophia has been shown to behave like signSGD on certain parameters, which prior studies [2,3] suggest may lead to suboptimal performance depending on the architecture.
>
> While developing a precise account behind the benefit remains a challenge, we hope the reviewer sees our contributions in this work given that the proposed method is rigorously evaluated across diverse under fair settings and sets a new state of the art.
>
> ---
>
> **Comparison with SAM” section**
>
> We apologize for the confusion by the current wording of the section.
>
> First, the main goal of Section 5.3 is to provide a controlled comparison between Sassha and SAM, demonstrating that Sassha is competitive with—or even outperforms—SAM. In line with this objective, we included an experiment in which SAM is given significantly more training iterations (x2) to show that Sassha still maintains superior performance, even under conditions that are favorable to SAM.
>
> The addressing of block Hessian heterogeneity was not intended as a central claim of Section 5.3, but rather as one plausible hypothesis —based on prior literature [4,5]—for why Sassha may perform better than SAM on Transformer-based architectures. We did not mean to position this as the primary claim for the observed performance difference.
>
> We appreciate the valuable feedback and will revise the writing in the final version.
>
> ---
>
> **On reported performances**
>
> > “SAM gets 79.1 acc with ResNet50 on ImageNet, while the authors report 76.3”
>
> This discrepancy is primarily due to the significantly larger number of training epochs used in [6] compared to ours (400 epochs in [6] vs. 90 in our setting). Additionally, we find differences in the experimental setup—such as the learning rate scheduler (cosine in [6] vs. multi-step in ours) and the use of label smoothing in [6]—this complicates a direct comparison of final accuracy.
>
> > “Sam gets 83.7 acc with WRN-28-10 on CIFAR100 ... However, the authors report 82.9”
>
> First, we would like to clarify that the SAM (with SGD as the base optimizer) result reported in our paper is 83.14%, not 82.9% as mentioned. Additionally, we were unable to find a reference for the 83.7% accuracy cited by the reviewer. If possible, we would appreciate it if you could share the source. In our own investigation, we found that the performance numbers reported in reference [7] are consistent with our results.
>
> Most importantly, we want to emphasize that all methods in our study were evaluated under an identical experimental setup to ensure fair comparison. Furthermore, we have prepared full configurations and code to reproduce all reported results, which will be released alongside the camera-ready version.
>
> &nbsp;
>
> **Reference**
>
> [1] Amari et al., When Does Preconditioning Help or Hurt Generalization? \
> [2] Liu et al., Sophia: A Scalable Stochastic Second-order Optimizer for Language Model Pre-training \
> [3] Karimireddy et al., Error feedback fixes sign sgd and other gradient compression schemes \
> [4] Zhang et al., Why Transformers Need Adam: A Hessian Perspective \
> [5] Ormaniec et al., What does it mean to be a transformer? insights from a theoretical Hessian analysis \
> [6] Foret et al., Sharpness-Aware Minimization for Efficiently Improving Generalization \
> [7] Wu et al., CR-SAM: Curvature Regularized Sharpness-Aware Minimization

---

### Official Review · Reviewer_Cfv3 · 2025-03-12

**Overall Recommendation:** 4

**Summary:**

This paper introduces SASSHA (Sharpness-aware Adaptive Second-order Optimization with Stable Hessian Approximation), a novel second-order optimization method designed to improve generalization performance. The authors investigate why approximate second-order methods tend to generalize poorly compared to first-order approaches, finding they converge to sharper minima. SASSHA addresses this by incorporating a sharpness minimization scheme similar to SAM into a second-order framework while stabilizing Hessian approximation with square-rooting and absolute value operations. The method also enables efficient lazy Hessian updates, reducing computational costs. Across image classification (CIFAR, ImageNet) and language tasks (pretraining and finetuning), SASSHA consistently achieves flatter minima and superior generalization compared to existing second-order methods and often outperforms first-order approaches including SGD and SAM.

**Claims And Evidence:**

The claims are well-supported through extensive empirical evidence. The authors clearly demonstrate that:

- Second-order methods converge to sharper minima (Table 1, Fig. 2)
- SASSHA achieves flatter minima across different sharpness metrics
- SASSHA consistently outperforms other methods in generalization (Tables 2, 3)
- Square-rooting and absolute value operations stabilize training (Figs. 4, 7)
- SASSHA is robust to label noise (Table 5)
- Lazy Hessian updates are effective due to lower Hessian sensitivity (Fig. 5)

The theoretical convergence analysis (Theorem 4.4) provides a sound foundation for the approach.

**Essential References Not Discussed:**

The paper has a thorough literature review covering most relevant work. The authors mention recent second-order methods and sharpness-aware optimization approaches comprehensively.

**Experimental Designs Or Analyses:**

The experimental design is robust and thoughtfully constructed:

- Multiple dataset sizes (CIFAR-10/100, ImageNet) and domains (vision, language)
- Several model architectures (ResNet variants, WideResNet, Vision Transformer, GPT1-mini, SqueezeBERT)
- Comprehensive hyperparameter tuning detailed in Appendix D
- Multiple random seeds with standard deviations reported
- Investigation of label noise robustness
- Detailed ablation studies for each component

**Methods And Evaluation Criteria:**

The experimental methodology is thorough and appropriate for evaluating optimization algorithms. The authors:

- Compare against relevant state-of-the-art methods (AdaHessian, Sophia-H, Shampoo, SGD, AdamW, SAM)
- Use diverse tasks (image classification, language modeling, finetuning)
- Conduct extensive hyperparameter tuning for fair comparison
- Evaluate across multiple metrics (validation accuracy, loss, sharpness metrics)
- Analyze robustness, stability, efficiency, and computational cost
- Provide ablation studies for each component of their method

**Other Comments Or Suggestions:**

- The figures effectively visualize the differences in loss landscapes
- The stability analysis provides valuable insights for practitioners
- The M-SASSHA variant offers an efficient alternative with competitive performance

**Other Strengths And Weaknesses:**

Strengths:

- Novel combination of sharpness minimization with second-order methods
- Strong empirical results across diverse tasks
- Effective stabilization techniques for Hessian approximation
- Computational efficiency through lazy Hessian updates
- Thorough analysis and ablation studies
- Robustness to label noise

Weaknesses:

- Limited theoretical justification for why square-rooting works better than alternatives
- Mostly empirical validation of design choices
- Additional tuning parameter $\rho$ (perturbation radius) compared to pure second-order methods
- Performance gains on language tasks are more modest than for vision tasks

**Questions For Authors:**

1- How does SASSHA's performance scale to larger models and datasets beyond those presented? Have you tried applying it to very large language models or transformer-based vision models?

2- The paper shows SASSHA outperforms SAM in most settings. Is there a theoretical explanation for why combining sharpness awareness with second-order information works better than either approach alone?

**Relation To Broader Scientific Literature:**

The paper effectively positions its contributions within the optimization literature. It builds upon:

- Sharpness-aware minimization (SAM) for generalization
- Approximate second-order methods (AdaHessian, Sophia-H)
- Stable Hessian approximation techniques
- Insights about flat minima and generalization

The authors clearly differentiate their approach from existing methods and provide comprehensive comparisons.

**Theoretical Claims:**

I reviewed the convergence analysis in Section 4.5 and the expanded proof in Appendix C. The convergence theorem (Theorem 4.4) follows standard optimization theory for adaptive methods, incorporating both the effects of the perturbation and diagonal Hessian preconditioner. The proof correctly uses smoothness conditions and perturbation bounds. The theoretical analysis is sound within the given assumptions.

---

> ### Author Rebuttal · Authors · 2025-04-01
>
> We sincerely thank the reviewer for thoroughly reviewing our work and giving us insightful and constructive feedback. While we address the raised questions below, we would be keen to engage in any further discussion as needed.
>
> ---
>
> > “How does SASSHA's performance scale to larger models and datasets beyond those presented?”
>
> We appreciate the reviewer’s suggestion. We provide the validation performance of Sassha on larger models (ViT-B-32) below.
>
> |VIT base|ImageNet|
> |-|-|
> |Metric|val. acc.|
> |AdamW|66.90|
> |SAM(AdamW)|69.18|
> |AdaHessian|66.96|
> |Sophia-H|64.26|
> |Sassha|**69.82**|
>
> We observe that Sassha outperforms both first- and second-order baselines. We will include this in the updated paper.
> Additionally, we plan to include larger-scale models such as ViT-L and GPT2-small in the camera-ready version.
>
> Note on configuration. For ViT-B, we used the same hyperparameters from ViT-S due to the limited time frame of the rebuttal.
>
> ---
>
> > “Is there a theoretical explanation for why combining sharpness awareness with second-order information works better than either approach alone?”
>
> We believe that the strong performance of SASSHA stems from the complementary benefits of combining sharpness-awareness and second-order information. Specifically:
> - The flatness induced by sharpness minimization has been shown—both theoretically and empirically—to improve generalization [1–5], and
> - The effectiveness of preconditioning based on second-order information in adapting to the ill-conditioned geometry of deep learning has been well established in theory [6–9]. More recently, it has also been shown that optimal preconditioners can potentially accelerate the decrease of the population risk [10].
>
> These advantages appear to act synergistically. In contrast, using either technique in isolation may introduce certain limitations (e.g., sharpness minimization alone may struggle with ill-conditioned geometry, while second-order methods alone may converge to sharp minima).
>
> ---
>
> > “Additional tuning parameter $\rho$ (perturbation radius) compared to pure second-order methods” as a weakness
>
> While it is true that Sassha requires $\rho$, this might not necessarily be a fair criticism since (1) pure second-order methods are not applicable for deep learning, and (2) approximate second-order methods require (i.e., the fair baselines for the purpose of this work) their own hyperparameters to mitigate various issues such as training instability; for instance, AdaHessian requires the spatial averaging block size, Sophia requires a clipping threshold, and Shampoo requires a damping factor. It is also worth noting that Sassha is found to be generally robust to the range of $\rho$ values commonly used for SAM.
>
>
> &nbsp;
>
> **Reference**
>
> [1] Jiang et al., Fantastic generalization measures and where to find them, ICML 2019. \
> [2] Tsuzuku et al., Normalized flat minima: Exploring scale invariant definition of flat minima for neural networks using pac-bayesian analysis, ICML, 2020. \
> [3] H Petzka et al., Relative Flatness and Generalization,  NeurIPS, 2021. \
> [4] Foret et al., Sharpness-Aware Minimization for Efficiently Improving Generalization, ICLR, 2021. \
> [5] Orvieto et al., Anticorrelated Noise Injection for Improved Generalization, ICML, 2022. \
> [6] Boyd et al., Convex Optimization, Cambridge University Press, 2004. \
> [7] Nocedal et al., Numerical Optimization, Springer, 2006. \
> [8] Bottou et al., Optimization Methods for Large-scale machine learning, SIAM Review, 2018. \
> [9] Jiang et al., How Does Adaptive Optimization Impact Local Neural Network Geometry?, NeurIPS, 2023. \
> [10] Amari et al., When Does Preconditioning Help or Hurt Generalization?, ICLR, 2021.

---

### Official Review · Reviewer_sYMy · 2025-03-17

**Overall Recommendation:** 3

**Summary:**

The paper introduces SASSHA, a second-order optimization method designed to enhance generalization by explicitly reducing the sharpness of minima through a sharpness-aware framework, while stabilizing Hessian approximations via techniques like square-rooting and absolute value transformations. It incorporates lazy Hessian updates to maintain efficiency and demonstrates robustness across diverse tasks, including image classification and language modeling. Empirical results show SASSHA consistently outperforms existing first- and second-order methods, achieving flatter minima and superior generalization performance.

**Claims And Evidence:**

In this paper, authors propose a novel second-order method designed to enhance generalization by explicitly reducing sharpness of the solution. There lacks of therotical analysis on why this mathod enhance generalization.

**Essential References Not Discussed:**

N.A.

**Experimental Designs Or Analyses:**

A more comprehensive ablation study on key hyper-parameters—such as learning rate and perturbation radius—is critical. These parameters likely have a significant impact on model performance, and their sensitivity could undermine the method's robustness if not rigorously analyzed. Without understanding how variations in these hyper-parameters affect results, the practicality and reliability of the approach in real-world scenarios remain questionable.

**Methods And Evaluation Criteria:**

While SASSHA demonstrates efficiency improvements (e.g., lazy Hessian updates), its evaluation focuses on small-sized models (e.g., ResNets, ViT-s, GPT1-mini). The computational and memory demands of second-order methods like SASSHA may still hinder scalability to modern billion-parameter architectures or distributed training scenarios, which are common in large-scale language/vision models.

**Other Comments Or Suggestions:**

Publicly releasing the full source code, accompanied by detailed implementation guidelines and hyper-parameter configurations, is essential to ensure transparency and reproducibility.

**Other Strengths And Weaknesses:**

weakness.

Though SASSHA reduces tuning compared to methods like SAM, it still introduces new hyperparameters (e.g., perturbation radius, Hessian update interval). The paper acknowledges that lazy updates require careful balancing to avoid performance degradation, suggesting sensitivity to configuration choices that could complicate real-world deployment.

**Questions For Authors:**

N.A.

**Relation To Broader Scientific Literature:**

The authors present a novel stabilization framework for Hessian approximations to mitigate loss landscape sharpness.

**Theoretical Claims:**

The convergence analysis (Section 4.5) assumes convexity and smoothness, which are unrealistic for non-convex neural networks. No theoretical guarantees connect sharpness minimization to generalization in practical settings.

---

> ### Author Rebuttal · Authors · 2025-04-01
>
> We really appreciate the reviewer taking the time to engage with our work. We address the specific points below and would be happy to clarify any remaining concerns.
>
> ---
>
> **Theory for improved generalization**
>
> The relationship between flatness and generalization is theoretically well-established, and many prior studies have shown that flat solutions correlate with better generalization performance [1-3].
> We are currently developing a theory to prove the improvement of Sassha's generalization. Precisely, we have obtained the result that Sassha finds a flat solution using linear stability analysis [4] (which we will add in the final paper) and plan to use this result and existing flatness-based generalization bounds [3] to establish a generalization theory for Sassha.
> If you have any other suggestions, please let us know—we will try our best to incorporate them.
>
> ---
>
> **Convergence analysis for non-convex neural networks**
>
> This is indeed a valid limitation. However, convergence analyses under such assumptions are quite standard in optimization literature [6-8]. Moreover, convergence analysis under more realistic conditions, such as non-smoothness, remains a challenge and is being actively studied even for standard deep learning optimizers like SGD and Adam [9].
>
> Nonetheless, we intend to analyze convergence properties of Sassha in non-convex settings. To achieve this, we plan to leverage frameworks for preconditioned optimizers that rely on Hessian eigenstructure analysis [10] and convergence analyses under mild assumptions in stochastic settings [11].
>
> ---
>
> **Scalability & Large-scale evaluations**
>
> The scalability of second-order optimization is a valid concern. However, despite such limitations, the broader community consensus is that the benefits provided by second-order methods—such as preconditioning—are potentially significant, and substantial collective efforts have made its computational complexities comparable to first-order methods[6, 12] ($O(n^3) \rightarrow O(n)$ in time complexity, and $O(n^2) \rightarrow O(n)$ in memory complexity).
>
> Nevertheless, more generally, it is quite natural that leveraging more information for performance improvement can entail increased computations, which is understood as a cost-performance tradeoff. Perhaps more importantly, what matters would be whether or not the tradeoff is reasonable compared to existing alternatives, and precisely in that sense, Sassha sets a solid stand among other second-order baselines.
>
> Also, we hope that the reviewer understands that the reason we have not evaluated billion-parameter scale models is not due to a fundamental limitation of second-order methods, but rather to enormous resource requirements to train models of such scale that we are unable to afford in our environments. In order to train an 8.3B parameter model using Adam, for instance, NVIDIA has employed 512 V100 GPUs (~16 TiB) [14].
>
> ---
>
> **Hyperparameters**
>
> Thank you for sharing your concern. However, we would like to note that the results for Sassha were obtained using values within standard ranges commonly adopted in prior work [4, 13, 17], and Sassha demonstrates strong performance without excessive tuning. We kindly refer you to Appendix D for detailed information on the hyperparameter search spaces across all experimental settings.
>
> Also, regarding the Hessian update interval, we have already shown that Sassha is extremely robust to this hyperparameter, as demonstrated in Fig. 5(a). Please find a more detailed discussion in Section 6.3.
>
> ---
>
> **Source code**
>
> We have prepared a source code that reproduces all results presented in the paper, and we plan to release it alongside the camera-ready version.
>
> &nbsp;
>
> **Reference**
>
> [1] Jiang et al., Fantastic generalization measures and where to find them \
> [2] H Petzka et al., Relative Flatness and Generalization \
> [3] Foret et al., Sharpness-Aware Minimization for Efficiently Improving Generalization \
> [4] Wu et al., how sgd selects the global minima in over-parameterized learning: a dynamical stability perspective \
> [5] Shin et al., Critical Influence of Overparameterization on Sharpness-aware Minimization \
> [6] Liu et al., Sophia: A Scalable Stochastic Second-order Optimizer for Language Model Pre-training \
> [7] Bottou et al., Optimization Methods for Large-Scale Machine Learning \
> [8] Reddi et al., On the Convergence of Adam and Beyond \
> [9] Li, Rakhlin & Jadbabaie, Convergence of Adam Under Relaxed Assumptions \
> [10] Doikov et al., Spectral Preconditioning for Gradient Methods on Graded Non-convex Functions \
> [11]  He et al., Convergence of Adam for Non-convex Objectives: Relaxed Hyperparameters and Non-ergodic Case \
> [12] Yao et al., ADAHESSIAN: An adaptive second order optimization for machine learning \
> [13] Gupta et al., Shampoo: Preconditioned Stochastic Tensor Optimization, ICML, 2018. \
> [14] Shoeybi et al., Megatron-LM: Training Multi-Billion Parameter Language Models Using Model Parallelism

---

### Decision · Program_Chairs · 2025-05-01

**Decision:**

Accept (poster)

**Comment:**

The paper proposes a second-order optimizer termed Sassha based on sharpness-aware minimization (SAM).
The author's show how to tackle SAM's min-max problem with curvature information and highlight additional challenges to stabilize the method due to the interaction of driving curvature to zero (flatness), while simultaneously using its inverse to pre-condition the gradient.
Empirically, the method is verified to find flatter minima than other second-order methods.

The paper's quality is good in terms of novelty, experimental design/depth, and presentation:

- The majority of reviewers found combining second-order optimization with SAM to be novel and I agree with this assessment.
- The paper is well written, figures and tables are clean and legible
- Most reviewers agreed that the paper contains extensive experiments, demonstrating that the proposed method reaches flatter minima than second-order competitors, justifying individual design decisions, such as square-rooting the curvature estimate to stabilize the algorithm, and providing sensitivity analyses to various hyper-parameters.
- During the discussion, the authors presented concrete plans for addressing criticisms expressed by the reviewers, such as strengthening the theoretical connections between flatness and generalization. One bigger concern brought up by one of the reviewers were the theoretical motivations for square-rooting the curvature that were rather vague. I believe the authors should follow the suggestions of Reviewer Wdix and justify this decision purely empirically.

After including the comments brought up by the reviewers, this paper is in good shape and I recommend acceptance.